# The geometry of representational drift in natural and artificial neural networks

**Kyle Aitken** *, **Marina Garrett**, **Shawn Olsen**, **Stefan Mihalas**

MindScope Program, Allen Institute, Seattle, Washington, United States of America

* kyle.aitken@alleninstitute.org

## Abstract

Neurons in sensory areas encode/represent stimuli. Surprisingly, recent studies have suggested that, even during persistent performance, these representations are not stable and change over the course of days and weeks. We examine stimulus representations from fluorescence recordings across hundreds of neurons in the visual cortex using in vivo two-photon calcium imaging and we corroborate previous studies finding that such representations change as experimental trials are repeated across days. This phenomenon has been termed "representational drift". In this study we geometrically characterize the properties of representational drift in the primary visual cortex of mice in two open datasets from the Allen Institute and propose a potential mechanism behind such drift. We observe representational drift both for passively presented stimuli, as well as for stimuli which are behaviorally relevant. Across experiments, the drift differs from in-session variance and most often occurs along directions that have the most in-class variance, leading to a significant turnover in the neurons used for a given representation. Interestingly, despite this significant change due to drift, linear classifiers trained to distinguish neuronal representations show little to no degradation in performance across days. The features we observe in the neural data are similar to properties of artificial neural networks where representations are updated by continual learning in the presence of dropout, i.e. a random masking of nodes/weights, but not other types of noise. Therefore, we conclude that a potential reason for the representational drift in biological networks is driven by an underlying dropout-like noise while continuously learning and that such a mechanism may be computational advantageous for the brain in the same way it is for artificial neural networks, e.g. preventing overfitting.

**Data Availability Statement:** All data used in this work is available at the Allen Brain Map, specifically https://portal.brain-map.org/explore/circuits/visual-coding-2p and https://portal.brain-map.org/explore/circuits/visual-behavior-2p for the Passive and Behavioral data, respectively. The code used in

## Author summary

Recently, it has been shown that the neuronal representations of sensory information in the brain can vary, even during seemingly stable performance. Why such "representational drift" occurs in the brain is currently unknown. In this work, using experimental data that images thousands of neurons across many mice, we precisely quantify how certain representations change over time with geometric tools used to understand high-dimensional data. Across two datasets where mice are either passively viewing a movie or actively performing a task, we find the representational changes have strikingly similar

this work can be found at: https://github.com/kaitken17/drift_geometry.

**Funding:** KA, MG, SO, and SM received funding from the Allen Institute. The funders had no role in study design, data collection and analysis, decision to publish, or preparation of the manuscript.

**Competing interests:** The authors have declared that no competing interests exist.

geometric properties. We then induce representational changes in an artificial neural network by injecting it with several distinct types of noise while it continues to adjust its components to maintain stable performance. Comparing the properties of its representational drift to what we observed in experiment, only a specific category of noise, known as "dropout", matches the geometry we observed in experiments. This hints at a potential biological mechanism underlying representational drift: a random suppression of certain neuronal components and a subsequent compensating change in other components. Additionally, dropout is well-known for helping artificial neural networks learn better, potentially hinting at a computational advantage to drift in the brain.

# 1 Introduction

The biological structure of the brain is constantly in flux. This occurs at both the molecular and cellular level, the latter through mechanisms such as synaptic turnover [1, 2]. For example, a subset of the boutons and side branches of axons in the primary visual cortex of adult Macaque monkeys were observed to appear/disappear on the timescale of several days [3] (though a subset of synaptic spines can be stable over much longer time scales [1]). Despite this significant turnover in the components, during adult life a healthy brain is able to maintain persistent performance and memory recall over timescales significantly greater than that of the biological changes. This has naturally led to the puzzle of how, once a task is learned or a long-term memory is stored, if the neuronal recording indeed represents that information [4, 5], how said representation changes over time without disrupting its associated function. Many recent studies have confirmed that, under persistent performance, neuronal encodings undergo "representational drift", i.e. a gradual change in the representation of certain information [6–12] (though see Refs. [13–15] for counterexamples).

This raises several questions about the nature of representational drift, that we will often call just "drift" throughout this work. To begin with, it is unclear how these representations change over time without a deterioration in performance. One potential mechanism that would be robust to such changes is that the brain encodes redundant representations. Redundancy that is robust to differences in neural activity has been observed in central pattern generating circuits of the brain [16]. Additionally, whether or not the brain's biological turnover is the cause of drift is also unknown. It has been suggested that there are computational advantages to drifting, and thus it may be something the brain has implemented as a tool for learning or memory [17, 18]. Finally, although studies have observed representational drift on time scales of minutes [11] to weeks [6–10, 12], the details of how drift changes as a function of time is also not clear.

It is our view that, in order to answer these questions regarding representational drift, it is important that we quantify drift's behavior by investigating its *geometric characteristics*. Such an analysis would allow us, for example, to construct more precise models of how drift occurs and better understand how it might change for different representations. More specifically, we would like to quantitatively define the neuronal representation of a given stimulus, understand how such a stimulus changes over time, and if such changes are at all dependent upon the geometry of the representations. If representations are characterized using vectors and subspaces of neural state space, the tools of geometry naturally arise in comparing such quantities across time and neurons. This leads us to perhaps more tractable queries such as how the magnitude of drift relates to the magnitude of the representation vector and whether or not there is any preferential direction to representational drift in neural state space.

As mentioned above, an additional benefit of further understanding the geometry behind drift is that it allows us to construct better models of how it occurs. With a further understanding of the nuances of drift's behavior, we can induce drift in models of the brain and look for what modifications need to occur in these systems to have drift with similar geometric characteristics to what we observe experimentally. For example, additional noise can be added to artificial neural networks (ANNs) in order to cause their feature spaces to drift. Exactly what type of noise is needed to match experimental observations can give us hints toward understanding drift's underlying mechanisms and perhaps its computational benefits.

Thus the goal of this work is to characterize the geometry of representational drift by studying how the neural state space representations change as a function of time. To this end, we study the feature space representations from in vivo 2-photon calcium imaging on the primary visual cortex from two experiments conducted on mice. Both datasets come from experiments that are conducted over several days, allowing us to understand how their feature space representations change. We find that the geometry of drift in the visual cortex is far from completely random, allowing us to compare these characteristics to drift in ANNs. We find drift in the two experimental paradigms resembles *dropout*-like noise in ANNs, i.e. a random masking of nodes/weights, tying these computational models back to the biological turnover observed in the brain.

**Contributions**. The primary contributions and findings of this work are as follows:

- To better understand how neuronal representations of mice change over time, we quantify said representations during both a passive viewing and an active behavioral visual task over a time-scale of days.

- When two neuronal measurements of excitatory cells in V1 are separated by timescales on the order of the days, we find that the change of neuronal activity due to drift is strongly biased toward directions in neural state space that are the most active (as measured by the variance/mean of $dF/F$ values).

- Representational drift occurs such that, on average, the most active neurons become *less* active at later time steps, indicating a bias toward representation turnover.

- We explore the presence of drift in the feature space of convolutional neural networks induced by several types of noise injected into the network and find the drift due to dropout, in particular *node* dropout [19], strongly resembles the geometrical properties of drift observed in experiments.

- We discuss how the resemblance of the experimental drift to the findings in artificial neural networks under dropout hints at both the mechanism behind drift and why drifting may be computationally advantageous, e.g. in helping to prevent overfitting.

**Related work**. Drift has been observed in the hippocampus [6, 7, 20, 21] and more recently in the posterior parietal [8, 22], olfactory [9], and visual [10–12] cortices of mice (see [18, 23, 24] for reviews). The timescale of many drift studies ranges from minutes to weeks. Notably, other studies have found stable representations over the course of months in the motor cortex and dorsolateral striatum of mice [13] as well as the forebrain of the adult zebra finch [14]. Despite a drift in the individual neurons, several studies have observed consistent population behavior across all days [6, 8, 9]. Representational drift is often thought of as a passive, noise-driven process. However, others have suggested it may be attributed to other ongoing processes in a subject's life including learning in which the influx of additional information requires a re-coding of previously learned representations [8, 24]. Studies have also observed differences in representational drift between natural and artificial stimuli, specifically it was

observed that there is significantly larger drift in natural movies than drifting gratings [10]. Finally, a recent study has found significant behavioral contributions to drift that occurs over the course of an imaging session, i.e. at the timescale of hours [25].

Representational drift has also been studied at the computational/theoretical levels [26–29]. In particular, Ref. [29] studies representational drift in Hebbian/anti-Hebbian network models where representations continually change due to noise injected into the weight updates. The authors find that the receptive fields learned by neurons drift in a coordinated manner and also that the drift is smallest for neurons whose receptive field response has the largest amplitude. Furthermore, they find that drift occurs in all dimensions of neural state space, suggesting that there is no subspace along which the network readouts might remain stable. Additionally, networks of leaky integrate-and-fire neurons with spontaneous synaptic turnover have observed persistent memory representations in the presence of drift [27]. The benefits of a geometrical analysis of neuronal representations has been the subject of a few recent works [30, 31].

**Outline**. We begin our geometric characterization of representational drift in experiments by discussing results from the Allen Brain Observatory [32], followed by the Allen Visual Behavior Two Photon dataset [33]. These datasets come from experiments where mice passively and actively view stimuli, respectively. Hence, throughout this work, these datasets will be referred to as the "passive data" and "behavioral data", respectively. We then follow this up by analyzing drift in artificial neural networks and show, under certain noise settings, its characteristics match the aforementioned experimental data. Finally, we discuss the implications of the similarity of representational drift in biological and artificial neural networks and how this relates to persistent performance and may be computationally advantageous. Additional details of our results and precise definitions of all geometric quantities are provided in the Methods section.

## 2 Results

### 2.1 Drift in passive data

In this section, we investigate the details of representational drift in the primary visual cortex over the time-scale of days from the Allen Brain Observatory dataset. We note that drift in this dataset was analyzed previously in Ref. [11] and we corroborate several of their results here for completeness.

**Experimental setup and neuronal response.** Over three sessions, mice are passively shown a battery of visual stimuli, consisting of gratings, sparse noise, images, and movies (Fig 1a). The neuronal responses in the visual cortex to said stimuli are recorded using in vivo 2-photon calcium imaging. We focus on the neuronal responses of one particular stimuli, "Natural Movie One", consisting of a 30-second natural image scene that is repeated 10 times in each of the three sessions. We analyze data from mice belonging to Cre lines with an excitatory target cell class that are imaged in the primary visual cortex. Crucially, a subset of the neurons imaged across the three sessions can be identified, allowing us to study how the neuronal response of said neurons changes across time. The time difference between the three sessions differs for each mouse and is at least one day.

To quantify the neuronal response, we divide Natural Movie One into 30 non-overlapping 1-second blocks. We define the $n$-dimensional *response vector* characterising the neuronal response to a given block as the time-average $dF/F$ value over the 1-second block for each of the $n$ neurons (Fig 1b, see Methods for additional details) [11]. Additionally, we define the response vector to only contain neurons that are identified in *all three* sessions. This will result in a conservative estimate of the amount of drift, as a cell which does not have any activity in

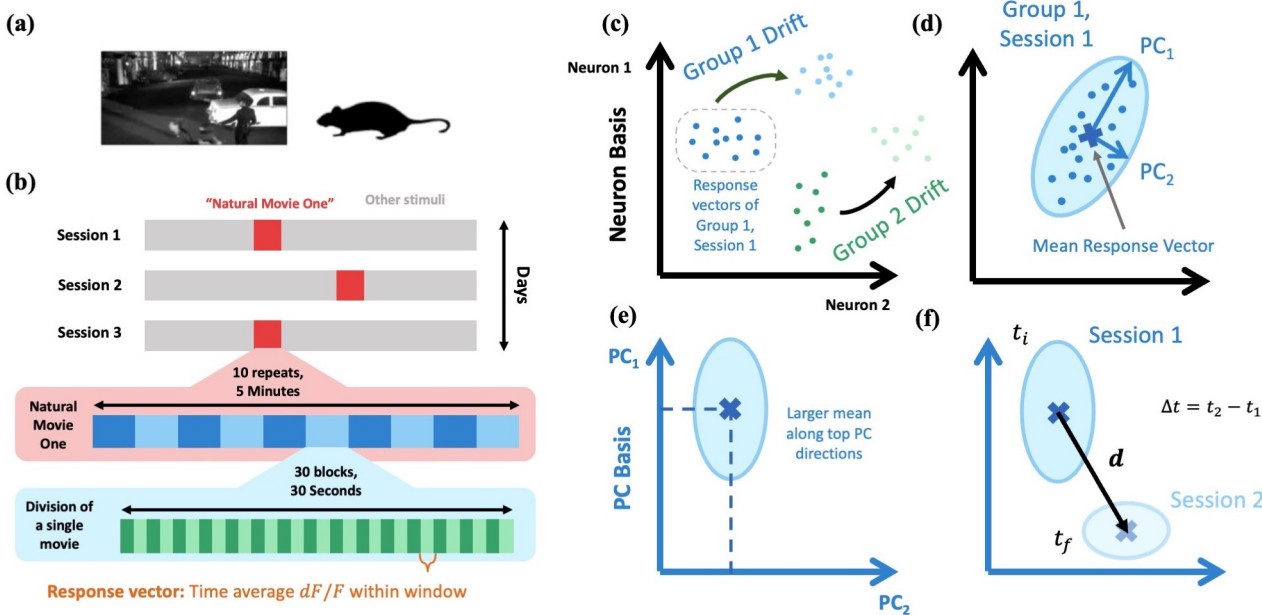

**Fig 1. Setup of passive data and visualization of feature space representations. (a)** Summary of passive data experiment. **(b)** Summary of response vector extraction across three imaging sessions. **[c-f]** Visualization of feature space representations. **(c)** Drift of response vectors belonging to two separate stimulus groups between two sessions. For example, stimulus group 1 might corresponding to the response vectors of the 0 to 1 second time-block of Natural Movie One and stimulus group 2 the 1 to 2 second time-block. **(d)** For each stimulus group in each session, we perform PCA to characterize the group's variation. An important quantity is also the group's mean response vector. **(e)** Moving to the respective stimulus group's PC basis, there is a strong correlation between the variance and mean value along a given direction, familiar from Poisson-like distributions. **(f)** The aforementioned feature space characterization is used to quantify drift. We define the drift vector, **d**, of a given stimulus group as pointing from the mean response vector of an earlier session (e.g. Session 1) to the mean response vector of a later session (e.g. Session 2). Δt is the time difference between the sessions.

an entire session can be missed by the segmentation. Throughout this work, we define the collection of response vectors corresponding to the same stimulus in a given session as the *stimulus group*, or just *group* for brevity. Thus, for the passive data, the response vectors of all 10 repetitions of a given time-block in a given session are members of the same stimulus group.

Between sessions, we will see the response vectors collectively drift (Fig 1c). To understand the geometry behind representational drift, we first quantify the feature space representations of the various stimulus groups in each session. The following quantification will be used on the passive data and throughout the rest of this work.

**Feature space geometry.** An important quantity for each stimulus group will be its *mean response vector*, defined as the mean over all $m$ members of a stimulus group in a given session (Fig 1d). Since there are 10 repetitions of the movie in each session, $m = 10$ for each stimulus group of the passive data. To characterize the distribution around this mean, for each stimulus group in each session, we perform principal component analysis (PCA) over all $m$ response vectors. This gives us the $PC_i$ directions, each of which we can associate with a *ratio of variance explained*, $0 \leq v_i \leq 1$, for $i = 1, \ldots, N$ and $N \equiv \min(m, n)$ the number of PCs for the particular stimulus group and session. The PC directions are ordered such that $v_i \geq v_j$ for $i < j$. We define the dimension of the feature space representation, $D$, by calculating the "participation ratio" of the resulting PCA variance explained vector,

$$D \equiv \frac{\left( \sum_{i=1}^{N} v_i \right)^2}{\sum_{i=1}^{N} v_i^2}, \tag{1}$$

where $1 \leq D \leq N$. $D$ thus quantifies roughly how many PC dimensions are needed to contain the majority of the stimulus group variance (Fig 1d). We define the *variational space* of a given stimulus group as the $\lceil D \rceil$-dimensional subspace spanned by the first $\lceil D \rceil$ PC vectors (where $\lceil \cdot \rceil$ is the ceiling function). Lastly, to eliminate the directional ambiguity of PC directions, we define a given PC direction such that the stimulus group has a positive mean along said direction.

Across all datasets analyzed in this work, we find the mean along a given PC direction is strongly correlated with its percentage of variance explained (Fig 1e). That is, directions which vary a lot tend to have larger mean values, familiar from Poisson-like distributions and consistent with previous results [34]. Below we will show how the above feature space representations allows us to quantify certain characteristics of drift between sessions (Fig 1f).

As mentioned above, each stimulus group from a given session of the passive data consists of 10 members, corresponding to the response vectors of a given 1-second block from 10 movie repeats (Fig 2a). Across stimulus groups, sessions, and mice ($n_{mice} = 73$), we find $D$ to be small relative to the size of the neural state space $D/n = 0.05 \pm 0.04$, but the variational space captures $91 \pm 4\%$ of the group's variation (mean $\pm$ s.e.) [35, 36]. Note that the fact that the variational space is relatively small and yet captures a large variation in the data is not too surprising given $m = 10$, i.e. there are only 10 members of each stimulus group, and hence $D$ could be at most 9. We find $D = 2.7 \pm 1.1$ (mean $\pm$ s.e.). Below we will show these numbers are consistent with larger stimulus groups. We also find the mean along a given PC direction is strongly correlated with its percentage of variance explained (S1 Fig).

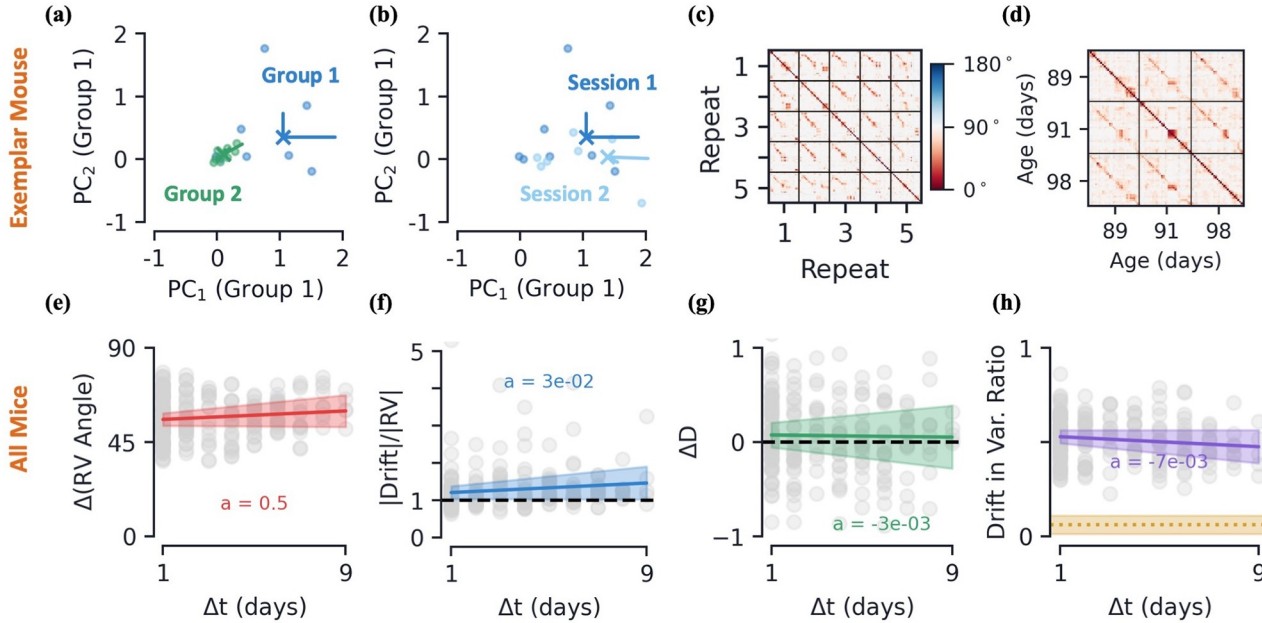

**Fig 2. Passive data: Feature space, drift, and drift's dependence on time. [a-d]** Data from an exemplar mouse. **(a)** Response vectors (dots), mean response vectors (X's), and first two PC dimensions (lines, scaled by variance explained), for two stimulus groups in a given session. Plotted in stimulus group 1's PC space. **(b)** Same as previous subplot, but response vectors of stimulus group 1 across two different sessions, plotted in session 1's PC space. **(c)** Pairwise angle between the response vectors of the 30 1-second time-blocks across the first five movie repeats of a single session. **(d)** Pairwise angle between mean response vectors across the three different sessions, same color scale as (c) (Methods). **[e-h]** Various metrics as a function of the time between earlier and later session, $\Delta t$, for all mice. All metrics are computed for each individual stimulus group, then averaged across all 30 groups. Colored curves are linear regression fits and shaded regions represent all fits within 95% confidence intervals of slope and intercept. **(e)** Average angle between mean response vectors. **(f)** Average (L2) magnitude of drift relative to magnitude of earlier session's mean response vector. **(g)** Average change in variational space dimension, $D$, from later session to earlier session. **(h)** Average drift magnitude within earlier session's variational space, ratio relative to full drift magnitude, see Eq (9). In yellow, the same metric if drift were randomly oriented in neural state space (mean $\pm$ s.e., across mice).

**Representational drift occurs between sessions.**   We now consider how the stimulus groups representing the 1-second blocks of Natural Movie One drift from one session to another (Fig 2b). Since we have three distinct sessions for each mouse, we can analyze three separate instances of drift, $1 \rightarrow 2$, $2 \rightarrow 3$, and $1 \rightarrow 3$.

We first verify the difference in neuronal representations between sessions is distinct from the within-session variation. To do so, we train a linear support vector classifier (SVC) to distinguish response vectors of a given stimulus group from one session to another session. We compare the 5-fold cross validated accuracy of this SVC to two SVCs trained to distinguish members of the same stimulus group within a *single* session. This is done by creating two types of within-session subgroups: (1) subgroups correspond to the first or second half of the session (2) subgroups correspond to either the even or odd movie repeats. The SVC trained to distinguish separate sessions achieves an accuracy significantly higher than chance (68 ± 8%, mean ± s.e.), while both within-session accuracies are at chance levels (51 ± 7 for first-second half, 45 ± 5% for even-odd, mean ± s.e.). We also note that previous work found the mean activity rates, number of active cells, pupil area, running speed, and gradual deterioration of neuronal activity/tuning do not explain the variation between sessions [11].

Now let us quantify how the response vectors change as a function of time. Throughout this work, we use the angle between response vectors as a measure of their similarity. Across repeats but within the same session, we find the angle between response vectors corresponding to the same time-block to generally be small, i.e. more similar, relative to those belonging to different blocks (Fig 2c). Comparing the mean response vectors of stimulus groups across sessions, we find a smaller angle between the same group relative to different groups, but it is evident that the neuronal representation of some groups is changing across sessions, as shown by the greater angle between their mean response vectors (Fig 2d).

**Drift has a weak dependence on the exact time difference between sessions.**   As a measure of the size and direction of drift, we define the drift vector, **d**, as the difference in the *mean* response vector of a given stimulus group from one session to another (Fig 1f, Methods). Additionally, we denote the time difference between pairs of imaging sessions by $\Delta t$. We will always take $\Delta t > 0$, so we refer to the individual sessions between which we are measuring drift as the *earlier* and *later* sessions.

Recall that the number of days between sessions is mouse-dependent. In order to compare aggregate data across mice, we would like to better understand how certain features of drift change as a function of $\Delta t$. To this end, we compare how several characteristics of the stimulus groups change as a function of time between sessions (Methods). We see a very modest increase in the average angle between mean response vectors as a function of $\Delta t$ (Fig 2e). This indicates that mean response vectors are, on average, only becoming slightly more dissimilar as a function of the time between sessions ($< 1$ degree/day). Many other geometric characteristics of the drift do not change considerably as a function of $\Delta t$ as well. We note here that this result does not mean that drift at shorter timescales does not accumulate over time, nor does it mean longer timescale drift would also exhibit such a weak dependence on $\Delta t$. See the Discussion for further consideration of how this result compares to other works.

We see the magnitude of the drift vector, **d**, is on average slightly larger than that of the mean response vector (Fig 2f). This not only indicates that size of drift on the time scale of days is quite large, but also that the size of drift does not seem to be increasing considerably with the time difference between sessions. Across $\Delta t$, we also see very little change in the variational space dimension, $D$, indicating the size of the variational space is not changing considerably (Fig 2g). As a measure of the direction of the drift relative to a stimulus group's variation space, we consider the ratio of the drift's magnitude that lies within the *earlier* session's variational space (Methods). Across $\Delta t$ values, we find this is quite steadily around 0.5, meaning

about half the drift vector's magnitude lies within the relatively small variational space (Fig 2h). This is significantly higher than if the drift vector were simply randomly oriented in neural state space (Fig 2h).

We take the above metrics to indicate that the drift characteristics of excitatory cells in V1 are fairly steady across the $\Delta t$ we investigate in this work, 1 to 9 days. Thus, in the remainder of this section we aggregate the data by session and across all $\Delta t$ (see S1 Fig for additional plots as a function of $\Delta t$).

**Drift's dependence on variational space.** Seeing that the drift direction lies primarily in the earlier session's variational space, next we aim to understand specifically what about said variational space determines the direction of drift. Since, by definition, the variational space is spanned the stimulus group's PC vectors, this provides a natural basis to understand the properties of drift. Looking at the magnitude of drift along the earlier session's PC direction as a function of the PC direction's variance explained ratio, $v_i$, we find that the magnitude of drift increases with the PC direction's variance explained (Fig 3a). That is, stimulus group directions that vary a lot tend to have the largest amount of drift. Additionally, there is a strong trend toward drift occurring at an angle *obtuse* to the top PC directions, i.e. those with high variance explained (Fig 3b). Said another way, drift has an increasingly large tendency to move in a direction opposite to PC directions with a large amount of variance Furthermore, we find

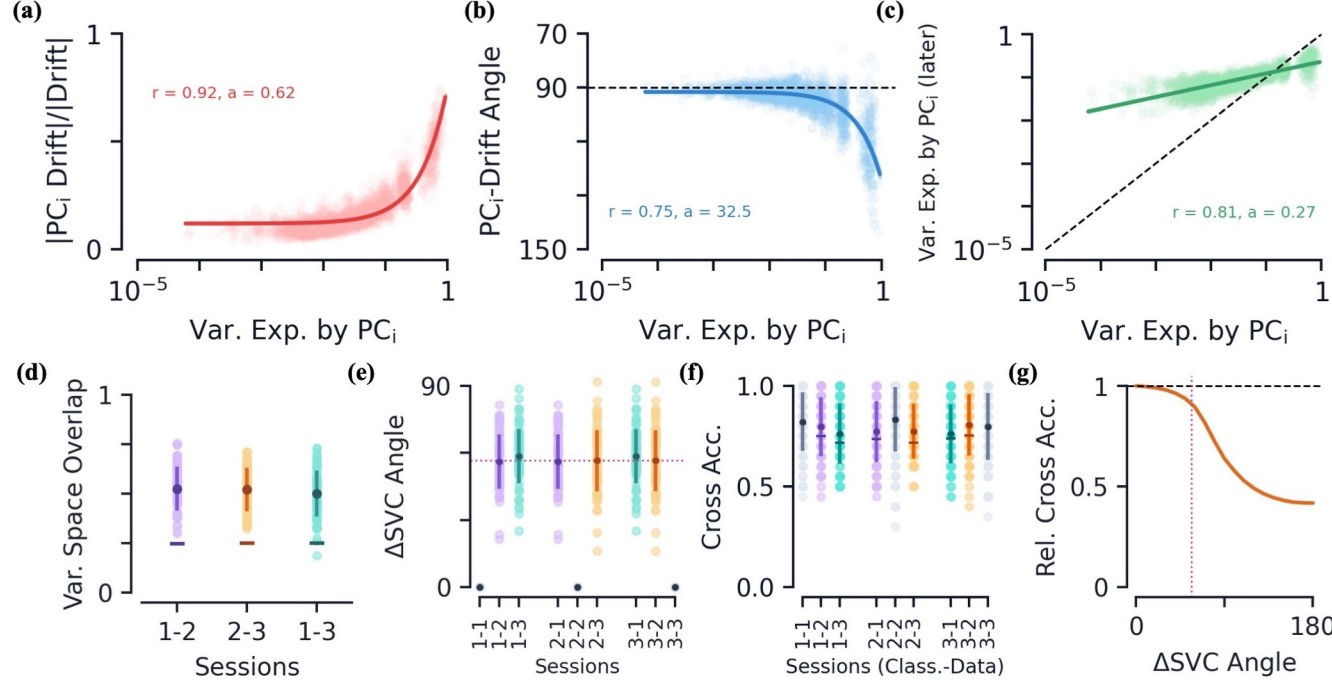

**Fig 3. Passive data: Drift geometry and classifier persistence.** [a-c] How various drift metrics depend on PC dimension of the *earlier* session's variational space. Metrics are plotted as a function of $PC_i$'s ratio of variance explained, $v_i$, across all stimulus groups. Colored curves are linear regression fits and shaded regions (often too small to see) are all fits within the 95% confidence intervals of slope and intercept. **(a)** Magnitude of drift along $PC_i$ direction relative to full (L2) magnitude of drift. **(b)** Angle of drift with respect to $PC_i$ direction. **(c)** Post-drift variance explained along $PC_i$ direction, black dotted line is equality. Linear regression fit to log(var. exp.). [d-f] Various metrics and how they change between sessions. The darker dots/lines always show mean value with error bars of ± s.e. The lighter color dots show data from individual mice. **(d)** The variational space overlap between earlier and later stimulus groups, $0 \leq \Gamma \leq 1$. The "−" marker indicates the average value of $\Gamma$ for randomly oriented variational spaces of the same dimensions. **(e)** Angle between linear support vector classifiers (normal vector) trained on distinct sessions. The purple dotted line is the average angle between different sessions. **(f)** Cross classification accuracy as a function of trained data session (Class.) and tested data session (Data). The "−" marker shows average classification accuracy when SVCs are randomly rotated by same angle that separates respective sessions' classifiers. **(g)** The relative cross accuracy, see Eq (14), as a function of the angle of a random SVC rotation. The purple dotted line is again the average angle found between drift sessions, also shown in (e).

both the magnitude and angular dependence of drift as a function of variance explained to be well fit by linear curves (Fig 3a and 3b).

What is the net effect of the above geometric characteristics? On average, a drift *opposite* the direction of the top PC directions results in a reduction of the mean value along said directions. Since a PC direction's magnitude and variation are correlated (S1 Fig), this also causes a reduction of the variation along said direction. This can be seen directly by plotting the variance explained along the earlier session's PC directions before and after the drift (Fig 3c). We see a decrease in variance explained in the top PC directions (below the diagonal), and an increase in variance explained for lower PC directions (above the diagonal). So at the same time variance is flowing out of the top PC directions, we find additional directions of variation grow, often directions that had smaller variation to begin with, compensating for the loss of mean/variance. Thus the net effect of this drift is to reduce the stimulus group variance along directions that already vary significantly within the group and grow variation along new directions.

A byproduct of this behavior is that the variational space of a stimulus group should change as it drifts. To quantitatively measure the change in variational spaces, we define the *variational space overlap*, $\Gamma$ (see Methods for precise definition). By definition, $0 \leq \Gamma \leq 1$, where $\Gamma = 0$ when the variational spaces of the earlier and later sessions are orthogonal and $\Gamma = 1$ when the variational spaces are the same (and when one space is a subspace of the other, see Methods). Between all sessions, we find $\Gamma \approx 0.5$, which is not far from $\Gamma$ values if the subspaces were simply randomly oriented, indicating that the variational space of a given stimulus group indeed changes quite a bit as it drifts (Fig 3d).

**Classifier persistence under drift.**   Above we showed that drift is both large and has a preference toward turning over directions with large variation, thus significantly changing a stimulus group's variational space. Intuitively, this is difficult to reconcile with previous results (and results later in this paper) that have observed mice performance remains persistent despite a large amount of drift [6, 8, 9]. To quantify the separability of stimulus groups and how this changes under drift, for each session we train a linear SVC to distinguish groups within said session using 10-fold cross validation. In particular, to avoid feature space similarity due to temporal correlation in the movie, we train our SVCs to distinguish the response vectors from the first and last 1-second blocks of a given session.

For a given mouse, we find the linear SVCs trained on its sessions are significantly different from one another, as measured by the angle between their normal vectors, on average 57 degrees (Fig 2e). However, despite this difference, we find that when we use one session's SVC to classify response data from a *different* session, the accuracy on the data does not fall significantly (Fig 2f). Interestingly, this appears to be a result of the high-dimensionality of the neural state space, as has been discussed previously [8]. Randomly rotating our SVCs in neural state space by the same angles we found between SVCs of different sessions, we achieve only slightly lower accuaracies (Fig 2f). Indeed, calculating the ratio of accuaracies as a function of the angle of the random rotation, we observe a monotonically decreasing function that is relatively stable up to the average angle we observed experimentally (Fig 2g). Note the relative cross accuracy is still finite at a change in the SVC angle of 180˚ because the SVC does not achieve 100% classification accuracy, so even with the weights flipped the SVC gets a non-trivial number of examples correct. We find it interesting that drift occurs such that the angle between SVCs is large yet not large enough to cause a significant dip in accuracy when used across sessions. Although they investigate drift in the parietal cortex, Ref. [8] also finds population decoding accuracies to be relatively stable across days for the part of the task that yields the high decoding accuracy. However, neural recordings at other parts of the tasks and smaller cell counts appear less stable. In the olfactory cortex, Ref. [9] finds a significant degradation in classification accuracy

over time, though their decoders are tasked with a much more difficult 8-way classification and smaller cell counts.

## 2.2 Drift in behavioral data

Now we corroborate our findings of drift geometry in a separate dataset, the Allen Visual Behavior Two Photon project ("behavioral data"), consisting of neuronal responses of mice tasked with detecting image changes [33].

**Experimental setup and neuronal response.** Mice are continually presented one of eight natural images and are trained to detect when the presented image changes by responding with a lick (Fig 4a). After reaching a certain performance threshold on a set of eight training images, their neuronal responses while performing the task are measured over several sessions using in vivo 2-photon calcium imaging. Specifically, their neuronal responses are first imaged over two "familiar" sessions, F1 and F2, in which they must detect changes on the set of eight *training* images. Afterwards, two additional "novel" imaging sessions, N1 and N2, are recorded where the mice are exposed to a *new* set of eight images but otherwise the task remains the same (Fig 4b, Methods). Similar to the passive data, the time difference between pairs of imaging sessions, F1-F2 or N1-N2, is on the order of several days, but differs for each mouse.

We will be particularly interested in the neuronal responses of a mouse's success and failures to identify an image change, which we refer to as "Hit" and "Miss" trials, respectively (see Methods for a parallel study of "Change" and "No Change" stimulus groups, which yields qualitatively similar results to what we present here). We once again form a *response vector* of the neuronal responses by averaging *dF/F* values across time windows for each neuron. The time window is chosen to be the 600 ms experimentally-defined "response window" after an image change occurs (Fig 4b, Methods). Once again, we define the response vector to only contain cells that are identified in both of the sessions that we wish to compare. Furthermore, to ensure a mouse's engagement with the task, we only analyze trials in which a mouse's running success rate is above a given threshold (Methods).

Since each session contains hundreds of image change trials, we have many members of the Hit and Miss stimulus groups. We once again find the dimension of the feature space representations, $D$, to be small relative to the size of the neural state space. Specifically, $D/n = 0.06 \pm 0.04$ and $0.06 \pm 0.05$, yet it captures a significant amount of variation in the data, $0.79 \pm 0.06$ and $0.80 \pm 0.06$, for the Hit group of the familiar ($n_{\text{mice}} = 28$) and novel ($n_{\text{mice}} = 23$) sessions, respectively (mean ± s.e.) [35, 36]. Additionally, we continue to observe a strong correlation between the mean along a given PC direction and its corresponding variance explained (S2 Fig).

**Drift geometry is qualitatively the same as the passive data.** Once again, we distinguish the the between-session variation due to drift from the inter-session variation by training linear SVCs. We again find the SVCs trained to distinguish sessions achieve an accuracy significantly higher than chance. For example, in the familiar session drift, the Hit stimulus groups can be distinguished with accuracy $74 \pm 11\%$ (mean ± s.e., novel data yields similar values). Meanwhile, the SVCs trained to distinguish the first/second half of trials and the even/odd trials within a single session do not do statistically better than chance (familiar Hit groups, first/second: $57 \pm 8\%$, even/odd: $52 \pm 6\%$, mean ± s.e.). Additional quality control checks were performed on this dataset to ensure behavioral differences and imaging variability across days could not explain the drift we observed (Secs. 4.3.1 and 4.3.2, S7, S8, and S9 Figs).

Similar to the passive data, the exact number of days between F1 and F2, as well as N1 and N2, differs for each mouse. Across $\Delta t$ values, we find many of the same drift characteristics

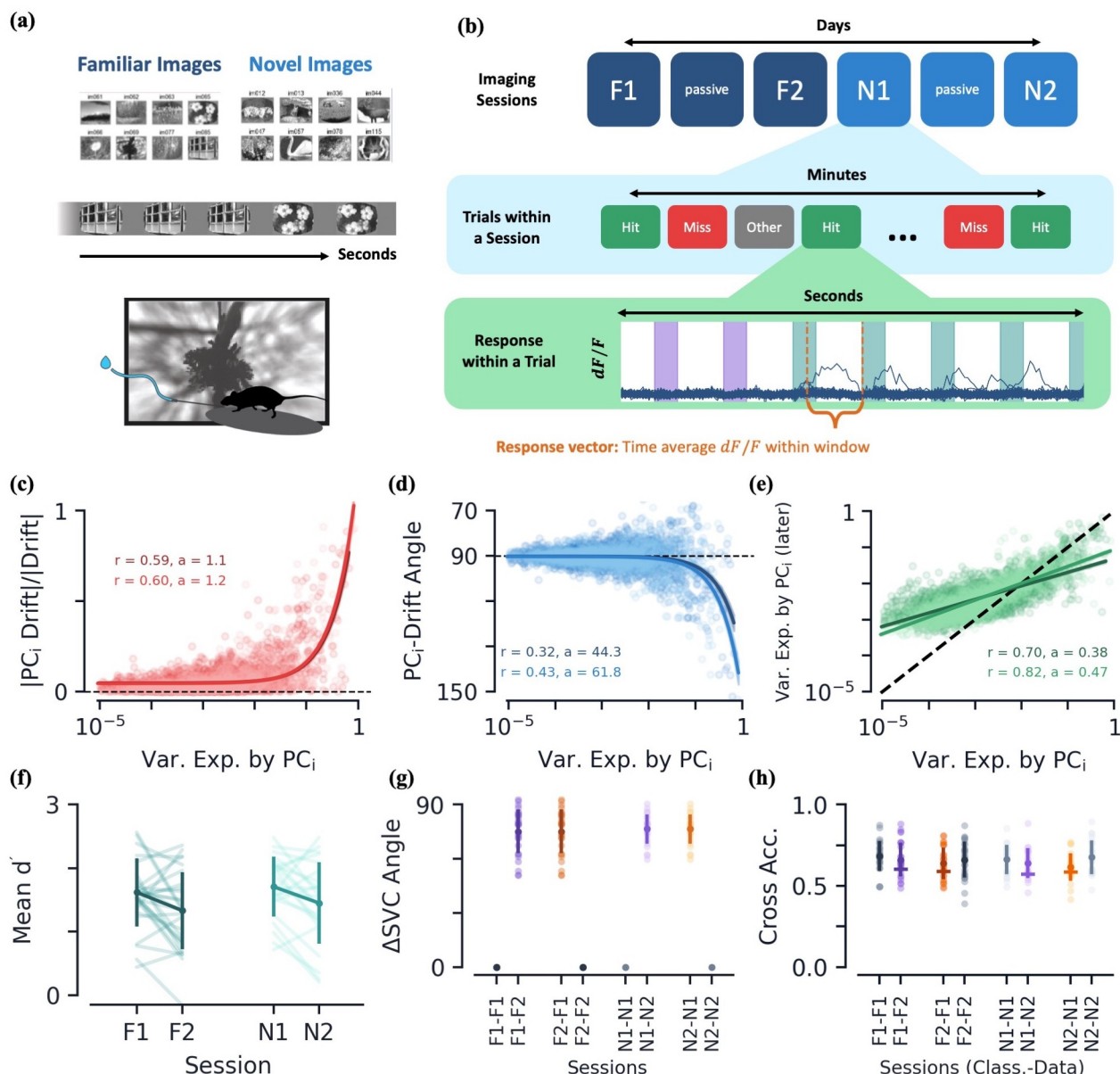

**Fig 4. Behavioral data: Experimental setup and drift geometry. (a)** Summary of experimental setup. **(b)** Summary of session ordering, trial types, and extraction of response vectors from $dF/F$ values. Bottom plot shows $dF/F$ values over time, with colored columns representing image flashes where different colors are different images. **[c-e]** Various drift metrics of Hit trials and their dependence on $PC_i$ direction of the earlier session's variational space. Dark colors correspond to drift between familiar sessions, while lighter colors are those between novel sessions. Metrics are plotted as a function of each $PC_i$'s ratio of variance explained, $v_i$. Colored curves are again linear regression fits. **(c)** Magnitude of drift along a given $PC_i$ direction, relative to full magnitude of drift. **(d)** Angle of drift with respect to $PC_i$ direction. **(e)** Post-drift variance explained along $PC_i$ direction (dotted line is equality). Linear regression fit to log(var. exp). **[f-h]** Various metrics as a function of session(s). Dark solid dots/lines show mean values with ± s.e. Light colored dots/lines show raw mice data. **(f)** Mean performance metric over engaged trails, $d'$ (Methods). **(g)** Angle between SVC normal vectors. **(h)** Cross classification accuracy, as a function of trained data session (Class.) and tested data session (Data). The "–" marker again shows average classification accuracy when SVCs are randomly rotated by same angle that separates respective sessions' classifiers.

we observed in the passive data, including: (1) a drift magnitude of the same order as the magnitude of the mean response vector, (2) on average, no change in the size of the variational space, and (3) a greater than chance percentage of the drift vector's magnitude lying within the earlier session's variation space (S2 Fig). Across all these measures, we do not find a significant

quantitative dependence on $\Delta t$, for the range of 1 to 6 days, so we will continue to treat drift data from different mice on equal footing, as we did for the passive data.

Between both the familiar and novel sessions, we again find the magnitude of drift along a given PC direction is strongly correlated with the amount of stimulus group variation in said direction (Fig 4c). Although their ratio is comparable to the familiar sessions, both the magnitude of drift and the mean response vectors are significantly larger in the novel sessions, consistent with previous findings (S2 Fig). Additionally, for both pairs of sessions, the drift again has a tendency to be directed *away from* the PC directions of largest variation (Fig 4d). The net effect of these characteristics is that we once again observe a flow of variation out of the top PC directions into directions that previously contained little variation (Fig 4e). It is fascinating that the familiarity/novelty of the image set does not seem to significantly affect the quantitative characteristics of these three measures.

**Inter-group characteristics under drift.**   Does the drift affect the mouse's performance? We observe no statistically significant difference in the performance of the mice despite the large amount of drift between sessions (Fig 4f). Notably, the novelty of the second image set does not affect the performance of the mice either, showing their ability to immediately generalize to the new examples.

We train a linear SVC to distinguish the Hit and Miss stimulus groups within a given session using 10-fold cross validation. Comparing the SVC between pairs of familiar/novel sessions, we again observe a significant amount of change between the SVCs as measured by the angle between their normal vectors (Fig 4g). Once again, an SVC trained on the earlier (later) session is able to classify the later (earlier) session's data with accuracy comparable to the classifier train on the data itself (Fig 4h). These are the same results we saw on the passive data: despite significant changes in the SVC, the stimulus groups do not seem to drift in such a way as to significantly change their linear separability.

One hypothesis for the persistence of performance under drift is that individual stimulus groups drift in a coordinated manner [11, 26, 29] (though some studies see a significant lack of coordination [9]). We find the average angle between the drift vectors of the Hit and Miss groups to be $68.5 \pm 16.5°$ and $56.9 \pm 14.6°$ for familiar and novel sessions, respectively (mean $\pm$ s.e.). That is, the drift directions are aligned at a level greater than chance (on average 90° for two random vectors), indicting that there is at least some level of coordination between the individual stimulus group drifts. Since we have found a tendency for drift to lie in the earlier session's variational space, an alignment in drift could be a byproduct of a similarity of the two groups' variational spaces. Indeed, we find the variational subspaces of the two stimulus groups to be aligned with one another at a rate significantly higher than chance, as measured by the variational space overlap, $\Gamma$ (S2 Fig).

## 2.3 Drift in artificial neural networks

Across two separate datasets observing mice passively or while performing a task, we have found qualitatively similar representational drift characteristics. We now turn to analyzing feature space drift in ANNs to try to understand what could possibly be the underlying mechanism behind this type of drift and its geometrical characteristics.

Convolutional neural networks (CNNs) have long been used as models to understand the sensory cortex (see [37] for a review). In this section, we analyze the effect of various types of noise on the feature space representations of simple CNNs, i.e. the node values of the penultimate layer (S4 Fig, Methods). Specifically, using the same geometrical analysis of the previous sections, we study how the variational spaces of different classes evolve as a function of time once the network has achieved a steady accuracy. If the feature space does drift, our goal is to

see if any of these types of noise cause the network's feature space to bear the qualitative properties we have found are present in representational drift the primary visual cortex analyzed in Secs. 2.1 and 2.2.

**Experimental setup.**  We train our CNNs on the CIFAR-10 dataset consisting of 60, 000 $32 \times 32$ color images from 10 different classes (e.g. birds, frogs, cars, etc.) [38]. Once a steady accuracy is achieved, we analyze the the time-evolution of the feature space representations under continued training of a two-class subset (results are not strongly dependent upon the particular subset). We take the analog of the *response vectors* of the previous sections to be the $n$-dimensional feature vector in the feature (penultimate) layer and the separate stimulus groups to be the classes of CIFAR-10. Throughout this section, we train all networks with stochastic gradient descent (SGD) at a constant learning rate and L2 regularization (Methods). Finally, we note that our goal in training the CNNs is to arrive at a feature space that somewhat resembles the neural data in the sense that the distinct groups, i.e. the CIFAR-10 classes, occupy somewhat distinct subspaces and the network's performance had stabilized. Specifically, we confirm the networks achieve stable accuracies well above chance, but do not optimize the networks' hyperparameters or learning schedules to achieve the highest possible accuracy for the given architecture.

**Different types of noise to induce drift.**  We begin by training our CNNs in a setting with minimal noise: the only element of stochasticity in the network's training is that due to batch sampling in SGD. Once our networks reach a steady accuracy, under continued training we observe very little drift in the feature space representations (red curve, Fig 5b). To induce feature space drift, we apply one of five types of additional noise to the feature layer of our networks:

1. **Additive node**: Randomness injected directly into the feature space by adding iid Gaussian noise, $\sim \mathcal{N}(0, \sigma^2)$, to each preactivation of the feature layer.

2. **Additive gradient**: Noise injected into the weight updates by adding iid Gaussian noise, $\sim \mathcal{N}(0, \sigma^2)$, to the gradients of the feature layer. Specifically, we add noise only to the gradients of the weights feeding *into* the feature layer. This is similar to how noise was injected into the Hebbian/anti-Hebbian networks studied in Ref. [29].

3. **Node dropout**: Each node in the feature layer is omitted from the network with probability $p$ [19, 39].

4. **Weight dropout**: Each weight feeding *into* the feature nodes is omitted from the network with probability $p$ [40].

5. **Multiplicative node**: Each feature node value is multiplied by iid Gaussian noise, $\sim \mathcal{N}(1, \sigma^2)$. This is also known as multiplicative Gaussian noise [39]. Note this type of noise is often seen as a generalization of node dropout, since instead of multiplying each node by $\sim$Bernoulli($p$), it multiplies by Gaussian noise.

All types of noise are applied both during initial training and the drift observation at steady accuracy. Of these five types of noise injection, below we will show that node dropout, and to a lesser extent multiplicative node and weight dropout, induce drift that strongly resembles that which we observed in both the passive and behavioral data.

Changes in the feature space are of course dependent upon the time difference over which said changes are measured. Similar to the experimental results above, below we will show that many of the drift metrics discussed in this section show no significant dependence upon $\Delta t$, so long as $\Delta t$ is large compared to the time scale of the noise. Additionally, the degree of drift

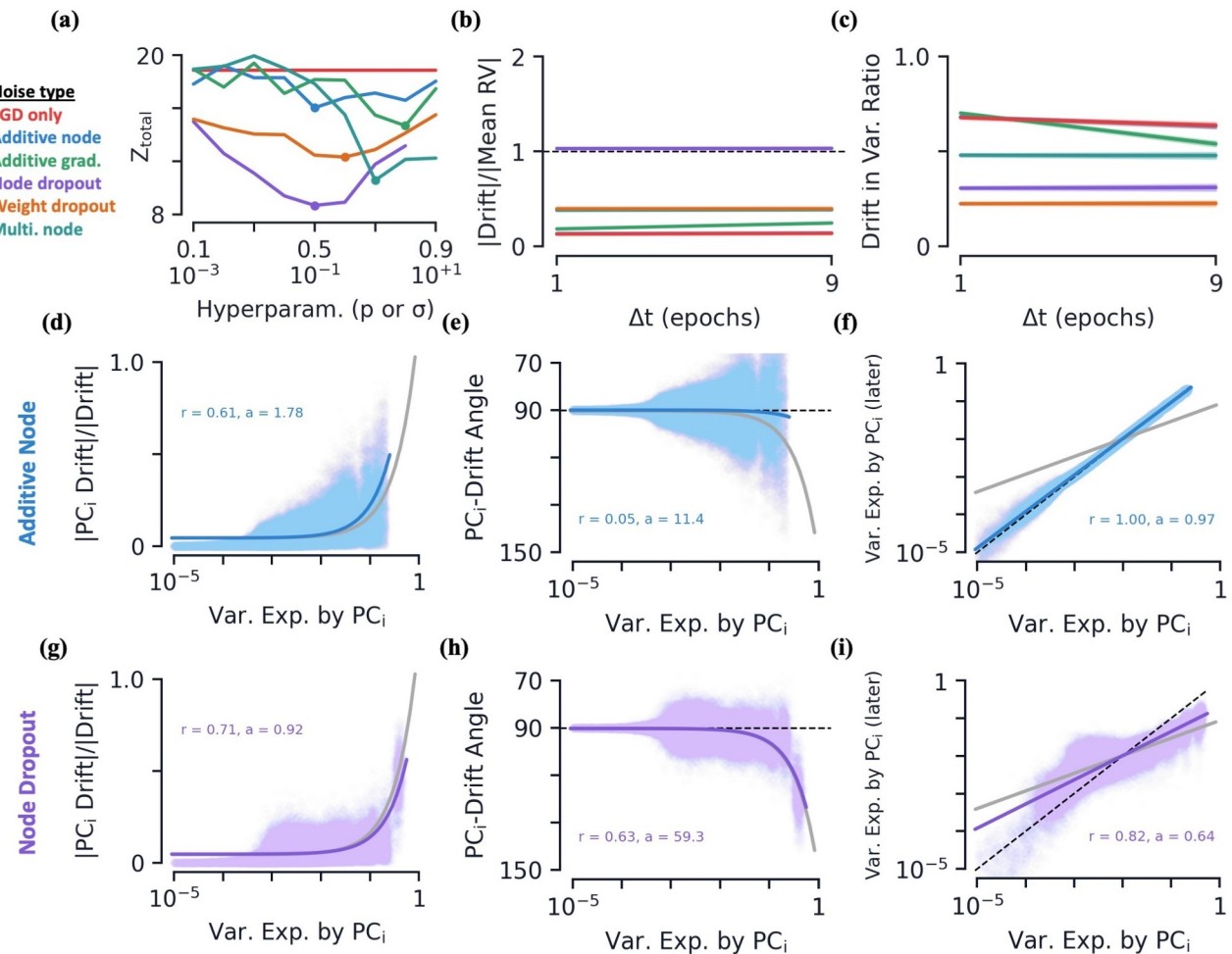

**Fig 5. Artificial neural networks: Hyperparameter fits and drift geometry as a function of Δ*t* and variance explained. (a)** Measure of fit to experimental data, $Z_{total}$ see Eq (19), as a function of noise hyperparameters, $p$ (top labels) or $\sigma$ (bottom labels). Dots are best fits, for which additional data is plotted here and in supplemental figures (S5 Fig, Methods). **[b-c]** Various metrics as a function of the time between earlier and later session, Δ*t*. Colored curves are linear regression fits. All data is averaged over 10 initializations. **(b)** Average magnitude of drift relative to magnitude of mean response vector. **(c)** Average percent of drift vector that lies in the variational space of initial session. **[d-i]** Various drift metrics and their dependence on PC dimension of the earlier session's variational space. Metrics are plotted as a function each $PC_i$'s ratio of variance explained, $v_i$, of the corresponding stimulus group Colored curves are linear regression fits. Grey curves are behavioral data fits from the novel sessions shown in Fig 4c, 4d and 4e. Middle row is for networks with additive Gaussian noise ($\sigma = 0.1$) and bottom row is with node dropout ($p = 0.5$). All data is averaged over 10 initializations. **(d, g)** Magnitude of drift along $PC_i$ direction, relative to full magnitude of drift. **(e, h)** Angle of drift with respect to $PC_i$ direction. **(f, i)** Post-drift variance explained along $PC_i$ direction (dotted line is equality). Linear regression fit to log(var. exp.).

found in the feature space of our CNNs is dependent upon the size of the noise injected, i.e. the exact values of $\sigma$ and $p$ above. For each type of noise, we conducted a hyperparameter search over values of $p$ (0.1 to 0.9) or $\sigma$ ($10^{-3}$ to $10^{+1}$) to find the values that best fit the experimental data (Fig 5a, Methods). Below we discuss results for the best fits of each types of noise. We find the qualitative results are not strongly dependent upon the exact values chosen, up to when too much noise is injected and the network does not train well.

**Feature space geometry and Δ*t* dependence.** Once more, we find the variational space of the various classes to be small relative to the neural state space. For example, under $p = 0.5$ node dropout we find $D/n = 0.070 \pm 0.003$, capturing $82.2 \pm 0.3\%$ of the variance. Notably, the feature space geometry continues to exhibit a correlation between variance explained and the

mean value along a given direction, again indicating that directions which vary a lot tend to have larger mean values (S4 Fig). Additionally, the variational spaces of the different classes continue to be aligned at a rate greater than chance (S4 Fig).

As expected, all five noise types are capable of inducing drift in the representations. This drift occurs amongst stable accuracies, that are comparable across all types of noise and steady as a function of time (S4 Fig). We find the size of drift relative to the size of the means to be comparable to that which was found in experiments for several types of noise (Fig 5b). Additionally, the relative magnitude of the drift for all types of noise is close to constant as a function of $\Delta t$. Similar to the experimental data, we find all the drifts do not induce a significant change in the dimensionality of the variational space (S4 Fig). Finally, we again note that the drift percentage that lies in variational space for all types of noise is significantly larger than chance, though all but the smallest drifts have a ratio smaller than that observed in experiment (Fig 5c).

Having observed several metrics that are constant in $\Delta t$, we use $\Delta t = 1/10$ epoch henceforth since it is within this weak $\Delta t$-dependence regime, and thus comparable to the experimental data analyzed in the previous sections. Decreasing the time scale of the noise injection to slower than each forward pass, it is possible to observe stronger $\Delta t$ dependence (Fig 6f). However, since our goal in this work is to match onto experimental data, where there we find evidence for very weak $\Delta t$ dependence, here we will focus on results within said regime.

**Dropout drift geometry resembles experimental data.** For clarity, here in the main text we only plot data/fits for the additive node and node dropout noises (Fig 5). Equivalent plots for SGD only and all other types of noise, as well as a plot with all six fits together, can be found in the supplemental figures (S5 Fig).

For all types of noise, we find an increasing amount of drift with PC dimension/variance explained, though the overall magnitude and distribution over variance explained vary with the type of noise (Fig 5d and 5g). All types of noise also exhibits a degree of increasing angle between the drift direction as a function of variance explained. However, for several types of noise, this trend is very weak compared to the experimental data, and it is clear the fitted data is qualitatively different from that observed in experiment (Fig 5e). The exceptions to this are node dropout, and to a lesser degree, weight dropout, where the fits and raw data match experiment quite well (Fig 5h). One way this can be seen quantitatively is by comparing the $r$-values of the linear fits, for which we see the node dropout data is relatively well approximated by the linear trend we also saw in the experimental data (S5 Fig). We see all types of noise result in a flow of variance out of the top PC dimensions and into lower dimensions. Once again though, for many types of noise, the amount of flow out of the top PC dimensions is very weak relative to the passive and behavioral data (Fig 5f). We do however see that the two types of dropout, as well as multiplicative node, all exhibit a strong flow of variation out of directions with large variations (Fig 5i and S5 Fig). Finally, it can also be helpful to compare drift geometry as a function of the earlier session's PC dimension instead of variance explained. See S6 Fig for plots of both experimental setups and all six ANN setups considered here.

From these results, we conclude that the three types of dropout, especially node dropout, exhibit features that are qualitatively similar to experiments. We now turn our focus to additional results under node dropout noise.

**Classifier and readouts persistence under drift.** Unique to the ANNs in this work, we have access to true readouts from the feature space to understand how the representations gets translated to the network's ultimate output. Previously, we fit SVCs to the feature space representation to understand how drift might affect performance, so to analyze our CNNs on an equal footing, we do the same here. Once again, we find classifiers fit at different time steps to be fairly misaligned from one another, on average 71 ($\pm$3) degrees (mean $\pm$ s.e., Fig 6a).

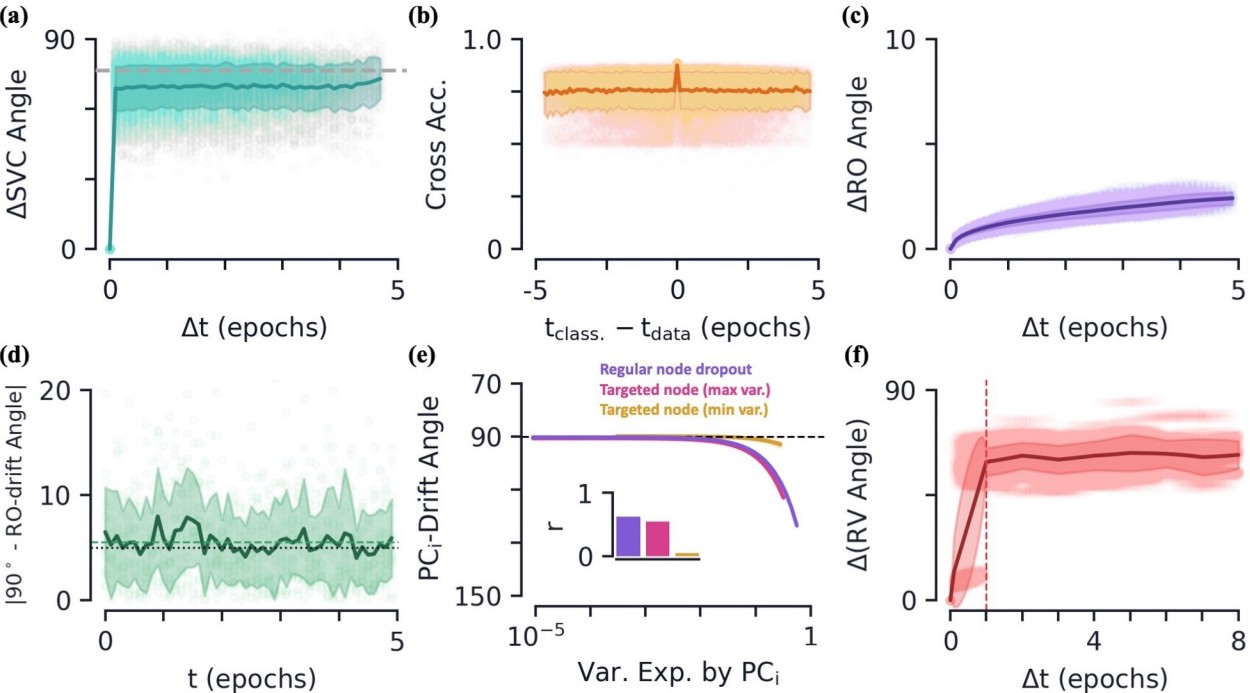

**Fig 6. Artificial neural networks: Additional properties of drift geometry.** [a–d] Various quantities as a function of relative/absolute training time (in epochs). Means are shown as dark lines, with 95% confidence intervals shaded behind. Raw data is scattered behind. **(a)** Angle between SVC classifiers (normal vectors) as a function of the time difference. The grey dashed line is the average for the novel Hit data shown in Fig 4g. **(b)** Cross classification accuracy as a function of time difference between classifier training time (Class.) and testing time (Data). **(c)** Difference in angle of a stimulus group's readout as a function of the time difference. Note the different vertical scale from (a). **(d)** Deviation of the angle between a stimulus group's drift and the respective readout from perpendicular (i.e. 90 degrees). The dashed green line is the average across time. The dotted black line is the angle between two randomly drawn vectors in a feature space of the same dimension. **(e)** Fits of variance explained versus angle of drift with respect to PC direction for regular node dropout (purple), targeted maximum variance node dropout (pink), and targeted minimum variance node dropout (yellow). The inset shows the $r$-values of the respective fits. **(f)** Difference in response vector angle as a function of $\Delta t$. The dashed vertical line indicates the time scale on which the node dropouts are updated (1/epoch).

Despite this, an SVC trained at one time step has slightly lower yet still comparable accuracy when used on feature space representations from another time step, with relative cross accuracy is 0.86 (±0.03) (Fig 6b). This is similar to what we observed in both the experimental datasets.

Interestingly, when we look at how the readouts of the CNN change with time, we see their direction changes very little, on average only 2.6 degrees over 5 epochs (Fig 6c). How can this be despite the large amount of drift present in the network? Comparing the direction of the drift to the network's readouts, we see that they are very close to perpendicular across time (Fig 6d). If the stimulus group means move perpendicular to the readouts then, on average, the readout value of the group remains unchanged. As such, despite the stimulus group drift being large, on average it does not change the classification results. Perhaps contradicting that this is a result of special design, we find the average angle between the drift and readouts to be consistent with chance, i.e. if the drift direction were simply drawn at random in the high-dimensional feature space. Thus we cannot rule out that the ability for the readout to almost remain constant in the presence of large drift is simply a result of the low probability of drift occurring in a direction that significantly changes the network's readout values in high dimensions [24]. Notably, we see comparatively more drift in the readouts for some other types of noise. For example, gradient noise causes a drift in the readouts of 18.6 degrees over 5 epochs.

**Additional drift properties in ANNs.**   Having established that the noise from node dropout strongly resembles that found in our earlier data, we now use this setting to gain some additional insights behind the potential mechanisms of representational drift.

Although we find our CNNs and experiments have comparable drift magnitude relative to the size of their mean response vectors, the CNNs appear to have significantly more variability due to drift compared to in-class variance than our experimental setups. For node dropout, we find SVCs trained to distinguish data from time steps separated by $\Delta t = 1/10$ epoch achieve perfect accuracy across trials, indicating the stimulus groups are linearly separable. SVCs trained to distinguish even/odd examples within a single class have chance accuracy, $49.3 \pm 0.8\%$ (mean $\pm$ s.e.), similar to experiment. The CNN also exhibits a coordination of drift between the two sub-groups of interest, whose drift vectors are separated by $40.4 \pm 4.9$ degrees (mean $\pm$ s.e.). As mentioned earlier, we also continue to observe a greater-than-chance variational space overlap between said stimulus groups (S4 Fig).

Next, we would like to see if we can further pinpoint what about node dropout causes the drift. To this end, we define a type of *targeted node dropout* algorithm that preferentially targets nodes with high variation (Methods). We find that qualitatively and quantitatively, targeted node dropout also has similar drift characteristics to the experimental data (Fig 6e and S4 Fig). Furthermore, this results holds for a smaller number of averaged nodes dropped per dropout pass, on average only 17 nodes per pass as compared to regular node dropout which drops $np = 42$ nodes per pass. Of course, with dropout percentages used in practice on the order of $p = 0.5$, the nodes that vary the most will be dropped quite frequently, so its not surprising that we are observing similar results here. If instead we target the nodes with the *smallest* variation, we do not observe a significant amount of drift or the characteristics we find in the experimental data, despite dropping the same number of nodes on average (yellow curve, Fig 6e and S4 Fig). Altogether, this suggests that it may be the dropping out of large variance/mean nodes that causes the characteristics of drift that we are observing.

In the above noise setups, we only injected additional noise in the feature layer, including the weights directly prior to said layer. We find that if dropout is also applied to earlier layers within the network, qualitatively similar drift characteristics continue to be observed (S4 Fig). However, when dropout was removed from the feature layer and applied only to an earlier layer, the amount of drift in the feature layer dropped significantly (S4 Fig).

In this work, we have focused on drift in the regime where certain metrics are close to constant as a function of $\Delta t$. As a means of verifying the transition into this regime, we can see if drift in ANNs is different on shorter time scales. To reach a regime where drift occurs slowly we lengthen the time scale of noise injection via node dropout by reducing the frequency of when the network recalculates which nodes are dropped, which is usually done every forward pass. When we do this, we observe a sharp transition in average angle between response vectors as a function of $\Delta t$ when it is above/below the noise-injection time scale (Fig 6f). We leave further observations of the differences in drift properties at such timescales for future work.

## 3 Discussion

In this work, we have geometrically characterized the gradual change of neuronal representations over the timescale of days in excitatory neurons of the primary visual cortex. Across experiments where mice observe images passively [32] and during an image change detection task [33], we observe similar geometric characteristics of drift. Namely, we find neuronal representations have a tendency to drift the most along directions opposite to those in which they have a large variance and positive mean. This behavior has the net effect of turning over directions in neural state space along which stimulus groups vary significantly, while keeping the

dimensionality of the representation stable. We then compared the experimentally observed drift to that found in convolutional neural networks trained to identify images. Noise was injected into these systems in six distinct ways. We found that *node* dropout, and to a lesser extent weight dropout, induce a drift in feature space that strongly resembles the drift we observed in both experiments.

Although weight dropout would qualitatively resemble the high noise observed at cortical synapses [41], it is interesting to speculate how the brain would induce node dropout in the primary visual cortex. Such an effect could arise in several different biologically plausible ways, including broad yet random inhibition across neurons. Such inhibition could potentially derive from chandelier cells, which broadly innervate excitatory neurons in the local cortical circuit. Importantly, chandelier cell connectivity is highly variable onto different pyramidal neurons [42]. Parvalbumin or somatostatin interneurons could also provide "blanket" yet variable inhibition onto excitatory cells [43]. Our findings that a targeted dropout of the most active artificial neurons induces a drift similar to uniform node dropout also suggests drift could come from an inhibition of the most active neurons, perhaps as an attempt to reduce the metabolic cost of the representation. Of course, the differences between node and weight dropout is simply a level of coordination of dropped weights. The equivalent of node dropout can also be achieved by either (1) dropping out all incoming weights and the bias of a given node or (2) all outgoing weights.

The ability for representational drift to occur during persistent performance was found in the behavioral task and is consistent with previous findings [8]. To understand how the separability of representations changes under drift, we have shown that when a linear SVC is trained to distinguish data at a given time, it can classify neuronal representations at some other time with comparable accuracy to the SVC trained on said data. This observation was found across both experiments and the artificial neural networks and suggests that drift in these systems occurs in such a way so as not to significantly change the representations' separability. In our ANN experiments, despite a significant amount of drift due to node dropout, we found the readouts remained relatively stable across time. Drift in these systems occurs very close to perpendicular to said readouts, though we did not find evidence that this was simply a result of the high-dimensionality of neural state space where two randomly drawn vectors are close to orthogonal. Nevertheless, the non-uniform geometric properties of drift we have observed do not rule out the possibility of a high-dimensional "coding null space" [24], which is different from other computational models where drift is observed to occur in all dimensions [29].

The resemblance of neuronal drift to that in artificial neural networks under dropout and continual learning suggests several computational benefits to the presence of such noise in the brain. It is a well known that dropout can help ANNs generalize as tool to avoid over-fitting and improve test accuracy, so much so that this simple modification has become commonplace in many modern ANN architectures. Most commonly, the benefits of dropout are linked to it approximately behaving as training using a large ensemble of networks with a relatively cheap computational cost [19]. That is, since dropout forces the network to learn many equivalent solutions to the same problem, the network's ultimate state is some amalgamation of the networks that came before it. Related to this, dropout can be interpreted as a Bayesian approximation to a Gaussian processes, suggesting it provides a robustness to overfitting because it essentially performs an "average" over networks/weights [44]. Indeed, it has been shown that redundant representations are found in the brain [16, 45] and such mechanisms may be why we can maintain persistent performance in the presence of drift [24].

In addition to the aforementioned effects, dropout has also been shown to have many other computational benefits in ANNs, many of which are easy to imagine might be useful for the brain. To name a few, dropout has been shown to prevent a co-adaptation of neurons [19];

impose a sparsification of weights [39]; be linked to weight regularization methods [46, 47]; be a means of performing data augmentation [48]; and, when performed with multiplicative Gaussian noise, facilitate the approximation of optimal representations [49]. From reinforcement learning, it is known that slow changes in activity are useful for gathering exploratory information and avoiding local minima [50].

Additionally, there are potential computational benefits of the turnover of components directly. Similar to the aforementioned results in ANNs, it has been shown there is a connection between plasticity in the brain and Bayesian inference [50]. Many theoretical models of the brain have explored the computational benefits of the turnover of components [50–52]. Additionally, overly rigid neural networks prevent learning and can also lead to catastrophic forgetting [53–55]. It has also been shown that cells which recently had large activity on the familiar dataset are more likely to be used for the novel dataset [8]. This suggests that drift may be computationally advantageous since it allows the mouse to have continuous performance while opening up new cells to learn new tasks/memories [8, 18, 27]. Finally, investigations of neuron turnover during memory updating have revealed mechanisms that seem to strongly resemble dropout-like behavior. In particular, the 'memory allocation hypothesis' postulates that neurons are predisposed to encode memories by their excitability, and the excitability of neurons is known to be modified by underlying biological fluctuations [18, 56–58]. In order to homeostatically maintain memory ensemble sizes, neurons must then also decrease their roles in the encoding [18], which could come from a depotentiation of competing memory ensembles [59, 60]. If this depotentiation targets neurons uniformly or even preferentially targets the most active neurons, this would mimic the dropout noise we injected into ANNs that produced a drift most similar to what we saw in the experimental datasets.

We have also found that many characteristics of drift vary quite slowly and may even remain constant over the time scales we have considered in this work (Fig 2 and S2 Fig). We note that this finding does not replicate other works that seem to see an accumulation in drift for certain periods of time. The lack of evidence in our study for time-dependence over these time-scales is not inconsistent with a slow time-dependence, as many of these other studies also seem to have evidence of drift eventually stabilizing [8, 10–12]. Almost all works use slightly different measures to determine similarity of representations and only a few have also studied drift in excitatory cells of V1 [10–12]. We also note the rate of exposure to other stimuli is different in the experimental datasets we examine in our work than those used in other works. For example, in the passive data we often have 3 exposures in $\leq 9$ days, with significant exposure to other stimuli during imaging sessions, the order of which change from session to session (Fig 1b). In the olfactory cortex, it has been shown that more frequent exposures to stimuli stabilize the representation [9], so the difference in stimuli exposure may be another reason our results differ from other works.

We also note that short timescale/within session drift has been observed in the same dataset in another study [11], and thus the variational space of a single session may contain contributions from said drift. We believe the fact that SVCs were able to distinguish across-session populations at much higher rates than within-session, time-separated populations shows that these contributions are small.

Finally we briefly highlight many open questions that arise from this work. Our study of drift has been limited to excitatory neurons in the primary visual cortex, it would of course be interesting to see if these geometric characteristics are present in other brain areas and neuron types, and if so, to understand if they quantitatively differ. Given that we have found the drift direction is stochastic yet far from uniformly random in neural state space, it would be beneficial to try to understand the manifold along which drift occurs and whether or not this acts as a "coding null space". Additionally, in this work we have limited our study to drift on the

timescale of days, and so this has left open a similar geometric understanding of short time-scale drift that has been observed in several studies and compare these results to the short time-scale drift we observed in ANNs. Lastly, given that we have found node dropout in ANN resembles the drift we found in experiment, it would be interesting to inject equivalent noise into more realistic network models of the brain, e.g. spiking neural networks [61], and see if the trend continues.

## 4 Methods

Here we discuss methods used in our paper in detail. Various details of the geometric measures we use throughout this work are given in Sec. 4.1. Further details of the passive and behavioral experiments and our analysis of the data are given in Secs. 4.2 and 4.3, respectively. Details of the artificial neural network experiments are given in Sec. 4.4. Supporting code for this work can be found at: https://github.com/kaitken17/drift_geometry.

### 4.1 Feature space and geometric measures

Throughout this section, we use $\mu, \nu = 1 \ldots, n$ to index components of vectors in the $n$-dimensional neural state space; $i, j = 1, \ldots, N$ to index the $n$-dimensional PC vectors that span a subspace of neural state space; and $I, J = 1, \ldots, \lceil D_p^s \rceil$ to index the PC vectors that span the variational space (see below for details).

**Feature and variational space.**  For stimulus group $p$ in session $s$, we define the number of members of said stimulus group to be $m_p^s$. Let the $n$-dimensional response vector be denoted by $\mathbf{x}_{p,\alpha}^s$ where $\alpha = 1, \ldots, m_p^s$ indexes members of said stimulus group. The *mean response vector* for stimulus group $p$ in session $s$ is then

$$\bar{\mathbf{x}}_p^s \equiv \frac{1}{m_p^s} \sum_{\alpha = 1, \ldots, m_p^s} \mathbf{x}_{p,\alpha}^s \,. \tag{2}$$

The dimension of PCA on stimulus group $p$ in session $s$ is $N_p^s \equiv \min(m_p^s, n)$. Denote the (unit-magnitude) $\text{PC}_i$ vector by $\mathbf{w}_{p,i}^s \in \mathbb{R}^{N_p^s}$ and the corresponding ratio of variance explained as $0 \leq v_{p,i}^s \leq 1$ for $i = 1, \ldots, N_p^s$ (alternatively, $\mathbf{w}_i$ and $v_i$ for brevity). PCs are ordered in the usual way, such that $v_i \geq v_j$ for $i < j$. We remove the directional ambiguity of the PC directions by defining the mean response vector of the corresponding stimulus group and session to have positive components along all PC directions. That is (suppressing explicit $s$ and $p$ dependence),

$$\bar{\mathbf{x}} \cdot \mathbf{w}_i = \frac{1}{m} \sum_{\alpha = 1, \ldots, m} \mathbf{x}_\alpha \cdot \mathbf{w}_i > 0 \,, \qquad \text{for } i = 1, \ldots, N. \tag{3}$$

If $\bar{\mathbf{x}} \cdot \mathbf{w}_i < 0$, then we simply redefine the PC vector to be $\mathbf{w}_i \to -\mathbf{w}_i$. The dimension of the feature space representation is the "participation ratio" of the PCA variance explained ratios

$$D_p^s \equiv \frac{\left( \sum_{i=1}^{N_p^s} v_{p,i}^s \right)^2}{\sum_{i=1}^{N_p^s} \left( v_{p,i}^s \right)^2} \,. \tag{4}$$

To build some intuition for this quantity, note that $D_p^s = N_p^s$ only when $v_{p,i}^s = 1/N_p^s$ for all $i = 1, \ldots, N_p^s$, i.e. the variance is evenly distributed amongst all $N_p^s$ PC dimensions. Additionally, $D_p^s = 1$ only when $v_{p,1}^s = 1$ and $v_{p,i}^s = 0$ for all $i > 1$, i.e. all the variance is in the first PC dimension. This measure is often used as a measure of subspace dimensionality [62–64]. Also

note this quantity is invariant up to an overall rescaling of all $v_i$, so it would be unchanged if one used the variance explained in each PC dimension instead of the *ratio* of variance explained. In general, this quantity is not an integer. Finally, the *variational space* of stimulus group $p$ in session $s$ is defined to be the $\mathbb{R}^{\lceil D_p^s \rceil}$ subspace spanned by the first $\lceil D_p^s \rceil$ PC vectors of said stimulus group and session, span $(\{\mathbf{w}_{p,I}^s\})$ for $I = 1, \ldots, \lceil D_p^s \rceil$, with $\lceil \cdot \rceil$ the ceiling function. As we showed in the main text, the dimension of the feature space, Eq (4), is often a small fraction of the neural state space, yet the variational space contains the majority of the variance explained.

We note that recently a similar quantification of feature spaces found success in analytically estimating error rates of few-shot learning [64].

**Drift between sessions.** We define the drift vector of stimulus group $p$ from session $s$ to session $s'$ to be

$$\mathbf{d}_p^{s,s'} \equiv \bar{\mathbf{x}}_p^{s'} - \bar{\mathbf{x}}_p^s. \tag{5}$$

Note that the ordering of indices in the superscript matter, they are (earlier session, later session), and will be important for later definitions. For this vector to be well-defined, $\bar{\mathbf{x}}_p^s$ and $\bar{\mathbf{x}}_p^{s'}$ must be the same dimension. This is always the case in the main text since we restrict to the subset of neurons that are identified in the sessions that we wish to compare. If the time of sessions $s$ and $s'$ are respectively $t^s$ and $t^{s'}$, we define the time difference between sessions $s$ and $s'$ by $\Delta t^{s,s'} \equiv t^{s'} - t^s$, dropping the superscripts in the main text. Note that if a neuron is not identified in a session it could either be because (1) the cell has left the imaging field or (2) the cell is inactive during the entire session and thus cannot be detected via calcium imaging. Although we wish to control for the former, the two cases were not distinguished in the datasets analyzed in this work and thus this methodology misses neurons that are completely inactive one session and active in another session.

We use the above geometric quantification for the passive data, active data, and artificial neural networks.

1. For the passive data, $p = 1, \ldots, 30$ corresponds to the non-overlapping one-second timeblocks of the movie. Meanwhile, $m_p^s = 10$ for all $p$ and $s$ since each session has ten movie repeats. Finally, $s = 1, 2, 3$ corresponds to the three different sessions over which each mouse is imaged.

2. In the behavioral data, we have $p =$ Hit, Miss and $s =$ F1, F2, N1, N2. In the supplemental figures, we also consider $p =$ Change, No Change, see Sec. 4.3 below for details (S3 Fig). The number of examples for each stimulus group is the number of engaged trials in a given session, so $m_p^s$ differs over both stimulus groups and sessions.

3. For the artificial neural networks, $p =$ cat, dog, car, $\ldots$, the distinct classes of CIFAR-10. In this case, $s$ represents the different time steps during training when the feature space measurements are taken. In this work we only consider $s$ values for which the the accuracy of the network is steady (see below for additional details). In practice, we use test sets where $m_p^s = 1000$ for all stimulus groups $p$.

**Geometric measures.** In this work, we often use the angle between two response vectors as a measure of similarity. This is different than the results of Ref. [11], where Pearson's correlation coefficient (see below) is used as a similarity measure between response vectors. Here we use angle because (1) our goal is to explore geometrical characteristics of drift, and we

believe angle is more interpretable in this context than Pearson's correlation and (2) angle is a rotationally invariant metric, and thus allows for comparisons independent of a mouse's particular basis of neural state space. Note neither of these measures are sensitive to the magnitude of the response vectors being compared. The two measures yield qualitatively similar results (S1 Fig).

The angle (in degrees) between two $n$-dimensional vectors $\mathbf{x}$ and $\mathbf{y}$ is defined as usual,

$$\theta(\mathbf{x}, \mathbf{y}) = \frac{180}{\pi} \arccos\left(\frac{\mathbf{x} \cdot \mathbf{y}}{\|\mathbf{x}\|_2 \|\mathbf{y}\|_2}\right) = \frac{180}{\pi} \arccos\left(\frac{\sum_{\mu=1}^{n} x_\mu y_\mu}{\sqrt{\left(\sum_{\mu=1}^{n} x_\mu^2\right)\left(\sum_{\nu=1}^{n} y_\nu^2\right)}}\right), \tag{6}$$

where $\|\cdot\|_2$ is the L2-normalization and $0 \leq \theta \leq 180$. Although the Pearson correlation coefficient is not used in this work, it is used in other works [11] as a quantitative measure of representation vector similarity. For the purpose of comparison, we reproduce the expression here,

$$r(\mathbf{x}, \mathbf{y}) = \frac{\sum_{\mu=1}^{n} (x_\mu - \bar{x})(y_\mu - \bar{y})}{\sqrt{\left(\sum_{\mu=1}^{n} (x_\mu - \bar{x})^2\right)\left(\sum_{\nu=1}^{n} (y_\nu - \bar{y})^2\right)}}, \tag{7}$$

where $-1 \leq r \leq 1$ and $\bar{x} \equiv \sum_\mu x_\mu$ and $\bar{y} \equiv \sum_\mu y_\mu$. From this expression we see the Pearson correlation coefficient is fairly similar to the angular measure, up the $\arccos(\cdot)$ mapping and the centering of vector components. Note neither $\theta$ nor $r$ are dependent upon the individual vector magnitudes, but the former is invariant under rotations of neural state space and the latter is not.

Often it will be useful to have a metric to quantify the relation between a drift vector to an entire subspace, namely a variational subspace. To this end, let the projection matrix into the variational subspace of stimulus group $p$ and session $s$ be $\mathbf{P}_p^s$. $\mathbf{P}_p^s$ can be constructed from the PCs of stimulus group $p$ in session $s$. Let $\mathbf{W}_p^s$ be the matrix constructed from the first $\lceil D_p^s \rceil$ (orthonormal) PCs of stimulus group $p$ in session $s$,

$$\mathbf{W}_p^s = \left[\mathbf{w}_{p,1}^s, \mathbf{w}_{p,2}^s, \cdots, \mathbf{w}_{p,\lceil D_p^s \rceil}^s\right]. \tag{8}$$

The column space of $\mathbf{W}_p^s$ is the variational space of stimulus group $p$ in session $s$. Then the projection matrix is $\mathbf{P}_p^s = \left(\mathbf{W}_p^s\right)^T$. The ratio of the drift vector that lies in the variational subspace of stimulus group $p$ in session $s$, or the *drift in variation ratio*, is defined to be

$$\gamma_p^{s,s'} = \gamma\left(\mathbf{P}_p^s, \mathbf{d}_p^{s,s'}\right) \equiv \frac{\|\mathbf{P}_p^s \mathbf{d}_p^{s,s'}\|_2^2}{\|\mathbf{d}_p^{s,s'}\|_2^2}, \tag{9a}$$

$$= \frac{1}{\|\mathbf{d}_p^{s,s'}\|_2^2} \sum_{I=1}^{\lceil D_p^s \rceil} \left(\mathbf{w}_{p,I}^s \cdot \mathbf{d}_p^{s,s'}\right)^2 \tag{9b}$$

where $0 \leq \gamma_p^s \leq 1$ and the second line follows from the fact the $\mathbf{w}_{p,I}^s$ form an orthonormal basis of the variational subspace. Intuitively, this quantity tells us how much of the drift vector lies in the variational space of stimulus group $p$. This is done by projecting the vector $\mathbf{d}$ into the subspace, and comparing the squared L2 magnitude of the projected vector to the original vector. If the drift vector lies entirely within the subspace, $\gamma_p^{s,s'} = 1$. Meanwhile, if the drift vector is orthogonal to the subspace, $\gamma_p^{s,s'} = 0$.

Finally, it will be useful to compare two variational subspaces to one another. To this end, we define the *variational space overlap* of stimulus group $p$ between sessions $s$ and $s'$ to be

$$\Gamma_p^{s,s'} \equiv \frac{1}{\min\left(\lceil D_p^s \rceil, \lceil D_p^{s'} \rceil\right)} \sum_{J=1}^{\lceil D_p^{s'} \rceil} \gamma\left(\mathbf{P}_p^s, \mathbf{w}_{p,J}^{s'}\right), \tag{10a}$$

$$= \frac{1}{\min\left(\lceil D_p^s \rceil, \lceil D_p^{s'} \rceil\right)} \sum_{I=1}^{\lceil D_p^s \rceil}\sum_{J=1}^{\lceil D_p^{s'} \rceil} (\mathbf{w}_{p,I}^s \cdot \mathbf{w}_{p,J}^{s'})^2, \tag{10b}$$

where $0 \leq \Gamma_p^{s,s'} \leq 1$ and in the second line we have used the fact the $\mathbf{w}_{p,I}^s$ are an orthonormal basis. Similar measures of subspace similarity are explored in Refs. [65, 66]. There it is also argued such measures are rotationally invariant to the orthonormal basis spanning either of the subspaces.

From the first line, we see $\Gamma_p^{s,s'}$ is simply a sum of the drift in variation ratio, Eq (9), of each basis vector of the variational space of session $s'$ relative to the variational space of $s$, weighted by the size of the *smaller* of the two variational spaces. From the second line, we see this measure is equivalent to a sum of the pairwise squared dot products between the PCs of the two variational subspaces, again weighted by the dimension of the smaller subspace. Additionally, from the second line it is clear that $\Gamma_p^{s,s'} = \Gamma_p^{s',s}$. It is straightforward to show $\Gamma_p^{s,s'} = 1$ if one variational space is a subspace of (or equal to) the other variational space. Additionally, $\Gamma_p^{s,s'} = 0$ if the two subspaces are orthogonal. As another example, if both variational subspaces were of dimension 6, and they shared an $\mathbb{R}^3$ subspace but were otherwise orthogonal, then $\Gamma_p^{s,s'} = 3/6 = 0.5$. In the S1 Appendix, we show this metric is also invariant under changes in the orthonormal basis spanning the subspaces. A quick way to argue this invariance is to notice in the definition of $\Gamma_p^{s,s'}$, Eq (10a), the magnitude of the projection of $\mathbf{w}_{p,J}^{s'}$ to the subspace $s$ is invariant under the rotation of the orthonormal basis of $s$. Additionally, from Eq (10b), $\Gamma_p^{s,s'}$ is symmetric with respect to the subspaces $s$ and $s'$, so if it is invariant under rotations of $s$ this must also be true for $s'$.

## 4.2 Passive data details

We refer to the Allen Brain Observatory dataset [32] as the "passive data" throughout this work. The dataset can be found at https://portal.brain-map.org/explore/circuits/visual-coding-2p along with significantly more details about its collection.

The passive data comes from experiments where mice are continually shown a battery of stimuli consisting of gratings, sparse noise, images, and movies over several sessions. Over three separate sessions, the neuronal response of the head-fixed mice is recorded using in vivo 2-photon calcium imaging. In this work we are solely concerned with the mices' response to the stimuli called "Natural Movie One", consisting of a 30 second natural-image, black and white clip from the movie *Touch of Evil* [67]. In each of the three sessions, the mouse is shown the 30-second clip 10 times in a row (five minutes total), for a total of 30 repeats across the three sessions. The sessions are separated by a variable time for each mouse, the vast majority of which are conducted on different days. Prior to the three imaging sessions, the mice were subject to a 2-week training procedure during which they were exposed to all visual stimuli to habituate them to the experimental setup.

**Table 1. Passive data Cre line details.**

| Cre line | Cortical layers | Number of mice |
|---|---|---|
| Emx1-IRES-Cre | 2/3, 4, 5 | 10 |
| Fezf2-CreER | 5 | 4 |
| Nr5a1-Cre | 4 | 7 |
| Ntsr1-Cre_GN220 | 6 | 5 |
| Rbp4-Cre_KL100 | 5 | 6 |
| Rorb-IRES2-Cre | 4 | 7 |
| Scnn1a-Tg3-Cre | 4 | 8 |
| Slc17a7-IRES2-Cre | 2/3, 4, 5 | 20 |
| Tlx3-Cre_PL56 | 5 | 6 |

Cre-lines, cortical layers, and number of mice for the passive data

In this work, we only analyzed mice from transgenic Cre lines where excitatory neurons were targeted (Table 1). The excitatory Cre line "Cux2-CreERT2" was omitted since it was observed to have several outliers compared to the rest of the lines. The cortical layers in which the fluorescence cells are present can change with Cre line, but are always within cortical layers 2/3, 4, 5, and/or 6. We omitted mouse data that had <10 shared cells amongst the three sessions (2 mice) as well as data where two sessions were taken on the same day (3 mice). In total, this left data from 73 mice for analysis. Sessions 1, 2, and 3 are temporally ordered so that $t_1 < t_2 < t_3$.

To verify between-session drift is distinct from within-session drift, we trained an SVC to distinguish members of a given stimulus group from one session to another or within the same session. That is, we labeled the response vectors of a given stimulus group by what sessions it is from for between-session drift. For within-session drift, we use two types of divisions to determine the labels of the individual response vectors, we either divide the trials into the first/second half of the session or by whether their trial number is even or odd. An SVC is then trained on each of these sets of response vectors and their labels using 5-fold cross validation. Accuracy data is then averaged across all mice and all stimulus groups. Decoder performance does not appear to improve with longer session gaps: for sessions 1 and 2 the SVCs were able to distinguish the sessions with 68 ± 9% accuracy, for 2 and 3 68 ± 8%, and for 1 and 3 69 ± 9% (mean ± s.e.).

For the plots as a function of $\Delta t$, we omit $\Delta t$ time scales where the data is too sparse. Specifically, we require each $\Delta t$ value to have at least 5 examples of drift. This results in the omission of some outlier $\Delta t$ values that are quite large (e.g. 22 days), but in total only removes 10 out of 219 distinct instances of drift, representing < 5% of the data.

Response vectors were averaged over 1-second blocks to allow for direct comparison to other work that analyzed drift in the same dataset [11].

**Fig 2 details.** Fig 2a was generated by projecting the response vectors for two stimulus groups onto the first two PC components of the first group. Similarly, Fig 2b was generated by projecting the response vectors for a given stimulus group from two different sessions onto the first two PC components of the group in the first session. Fig 2c simply consists of the pairwise angle between the response vectors of the first five movie repeats for an exemplar mouse. Since each repeat consists of 30 response vectors, in total there are $(5 \times 30)^2$ points shown. Fig 2d is generated by first finding the mean response vector, $\bar{\mathbf{x}}_p^s$, for each stimulus group $p = 1, \ldots, 30$. We then compute the pairwise angle within the session and between sessions, $\theta(\bar{\mathbf{x}}_p^s, \bar{\mathbf{x}}_p^{s'})$ for $s, s' = 1, 2, 3$.

Fig 2e was generated using the same mean response vectors for each stimulus group, where now the pairwise angle between the mean response vectors belonging to the same group are computed. This is done for all 30 stimulus group and then the average of all between-session angles is computed, i.e.

$$\bar{\theta}^{s,s'} \equiv \frac{1}{30} \sum_{p=1}^{30} \theta(\bar{\mathbf{x}}_p^s, \bar{\mathbf{x}}_p^{s'}), \qquad s \neq s'. \tag{11}$$

This quantity is plotted with its corresponding $\Delta t$ and then fitted using linear regression. Error bars for all linear regression plots throughout this work are generated by computing the 95% confidence intervals for the slope and intercept separately, and then filling in all linear fits that lie between the extreme ends of said intervals. Fig 2f, 2g and 2h are all computed similarly, the drift metric for a given division is computed and then, for each mouse and separate instance of drift, the quantity is averaged over all 30 stimulus groups. In particular, the quantities plotted in Fig 2f, 2g and 2h are respectively

$$\bar{d}^{s,s'} \equiv \frac{1}{30} \sum_{p=1}^{30} \frac{\|\mathbf{d}_p^{s,s'}\|_2}{\|\bar{\mathbf{x}}_p^s\|_2}, \tag{12a}$$

$$\overline{\Delta D}^{s,s'} \equiv \frac{1}{30} \sum_{p=1}^{30} \left( D_p^{s'} - D_p^s \right), \tag{12b}$$

$$\bar{\gamma}^{s,s'} \equiv \frac{1}{30} \sum_{p=1}^{30} \gamma_p^{s,s'}, \tag{12c}$$

as a function of $\Delta t$ between the earlier session $s$ and later session $s'$, see Sec. 4.1 above for definitions of the quantities on the right-hand side of these expressions. Note the distinction of $s$ and $s'$ matter for these quantities. Finally, for Fig 2h the chance percentage for drift randomly oriented in neural state space was computed analytically. For a randomly oriented drift vector projected onto a $\lceil D_p^s \rceil$ dimensional subspace, $\gamma_p^{s,s'} = \lceil D_p^s \rceil / n$. This quantity was averaged over stimulus groups and the three distinct drifts, before the average and standard error was computed across mice.

**Fig 3 details.** The scatter plots in Fig 3a, 3b and 3c are averaged over $p$ and plotted for each mouse and all three session-to-session drifts, i.e. $1 \rightarrow 2$, $2 \rightarrow 3$, and $1 \rightarrow 3$, as a function of the corresponding $\nu_i$. Any plot data with $\nu_i \leq 10^{-5}$ was omitted. For Fig 3a, 3b and 3c, they are respectively

$$d_i^{s,s'} \equiv \frac{1}{30} \sum_{p=1}^{30} \mathbf{d}_p^{s,s'} \cdot \mathbf{w}_{p,i}^s, \tag{13a}$$

$$\bar{\theta}_i^s \equiv \frac{1}{30} \sum_{p=1}^{30} \theta\left( \mathbf{d}_p^{s,s'}, \mathbf{w}_{p,i}^s \right), \tag{13b}$$

$$\bar{\nu}_i^{s,s'} \equiv \frac{1}{30} \sum_{p=1}^{30} \tilde{\nu}_{p,i}^{s,s'}, \tag{13c}$$

where $\tilde{\nu}_{p,i}^{s,s'}$ is the percent of variance explained of stimulus group $p$ in session $s'$ along the $i$th

PC dimension of stimulus group $p$ in session $s$ (i.e., $\mathbf{w}_{p,i}^{s}$). We found linear fits on the variance explained versus drift magnitude and PC-drift angle were better using a linear variance explained scale versus logarithmic variance explained scale.

For Fig 3d, we computed the $\Gamma_{p}^{s,s'}$, Eq (1), between the pair of variational spaces of the three possible pairs of sessions. This was done for each stimulus group $p$ and mouse, then the raw mouse data for each pair of sessions is the average value of $\Gamma_{p}^{s,s'}$ across all $p$. The average and $\pm$ s.e. bars were then computed across all mice. To generate the data in Fig 3e and 3f, we trained a linear SVC to distinguish the stimulus groups belonging to the first and last 1-second time-blocks of the movie. The first and last 1-second time-blocks were chosen to avoid any temporal correlation in the similarity of the movie scenes. For each mouse and each session, we trained our SVCs using 10-fold cross validation with an L2 penalty. The SVCs achieve an accuracy well above chance, 79 $\pm$ 15% (mean $\pm$ s.e.). For all folds belonging to sessions $s$ and $s'$, pairwise angles between their normal vectors were determined using Eq (6) before averaging over all 10 folds.

Finally, we wanted to investigate if there was something special about the orientation of the SVCs between sessions that allowed them to have large angular separation by still high cross-classification accuracy. To this end, we tested their accuracy relative to random rotations of SVCs. For a given pair of sessions $s$ and $s'$, we found the angle between the normal vectors of their respective SVCs. Using said angle, we generated a random rotation direction in neural state space and rotated the weights of the SVC of session $s'$ by the same angle. Let the accuracy of the SVC trained on data from session $s$, classifying the data from session $s'$, be denoted by $a^{s,s'}$. We define the *relative cross correlation accuracy* to be the accuracy of the cross-classification relative to the accuracy of classifier trained on the data, i.e.

$$a_{\text{rel.}}^{s,s'} \equiv \frac{a^{s,s'}}{a^{s',s'}} . \tag{14}$$

By definition, $a_{\text{rel.}}^{s,s} = 1$. We computed the relative cross correlation accuracy (again using 10-fold cross validation) of the randomly rotated SVC relative to that of session $s'$ and did this for 10 separate random rotations (where the angle is fixed, but the rotation direction varies). In practice, we found these randomly rotated SVCs only had slightly lower accuracy than the SVCs actually trained on the separate sessions (shown as "–" marks in Fig 3f). The plot shown in Fig 3g was generated in the exact same manner of randomly rotating SVCs, but now the angle is a set value instead of it being equal to the angle between respective sessions' SVCs. Note the cross-classification accuracy does not drop to 0 when the SVC is rotated 180 degrees (i.e. completely flipped) because, in general, the SVCs do not achieve 100% classification accuracy, so even the flipped SVC gets some examples correct (additionally, we are not changing the bias term).

## 4.3 Behavioral data details

Significantly more details on this dataset can be found in the technical whitepaper, see Ref. [33]. Here we provide a brief experimental overview for completeness. The dataset can be found at https://portal.brain-map.org/explore/circuits/visual-behavior-2p.

A mouse continually shown images from a set of eight natural images. Each sessions consist of several trials in which the flashed images change once or remain the same. Each trial consists of several 750 ms time-blocks that start with a single image shown for for 250 ms followed by grey screen for 500 ms (Fig 4b). The same image is shown several times at the start of a trial before an image change may occur. Specifically, for a trial in which the image changes, the

image change occurs between 4 and 11 time-blocks (equivalently, image flashes) into the trial. The number of flashes before a change are drawn from a Poisson distribution, so an image change after only 4 flashes occurs most frequently.

The general session schedule for in vivo 2-photon imaging consists of two active sessions separated by one passive session. In the passive sessions, whose data is omitted from this work, the mouse is automatically rewarded during any image change. After two active sessions and one passive session, the mouse is shown a novel set of eight images, under the same conditions. Notably, the set of eight "familiar" and "novel" images can be flipped for different mice.

Hit trials consist of trials where a mouse correctly responds to a lick within the "response window", between 150ms and 750ms after an image change. Miss trials are also image change trials, but where the mouse does not correctly respond. Across all sessions, we filter for mouse engagement, which is defined in the experimental white paper [33] to be a rolling average of 2 rewards per minute. Any trials (Hit or Miss) in which the mouse is not engaged are omitted. See Table 2 for a summary of trial counts.

As mentioned in the main text, to construct response vectors, neuronal responses are collected as mean vectors within the window between 150ms and 750ms after an image change. This window was chosen so as to match the "response window" where the mouse can respond with a lick [33]. Since the image flashes last 250ms, this window generally captures the mouse's neuronal response to both the image change as well as the reaction to the image turning off [33]. For each of the $n$ cells imaged in the given session, we average the $dF/F$ values in this time window to construct an $n$-dimensional response vector representing the neuronal response to the given trial.

A subset of the mice were imaged using multiplane imaging techniques, allowing excitatory cells outside of the primary visual cortex to also be imaged. For such mice, we only included data from cells in the primary visual cortex.

Mice included in both the feature and novel data were filtered under several quality control conditions. Several mice had testing schedules that did not match that pattern shown in Fig 4b, namely sessions in the order: F1, familiar passive, F2, N1, novel passive, N2. We omitted mice where a familiar/novel session was missing in either one of the paired imaging sessions. Occasionally, mice had no passive session separating either F1 and F2 or N1 and N2, but said mice were not omitted. Finally, we filtered for a minimum number of engaged trials, $\geq 10$ (across both Hit and Miss trials), and also a minimum number of shared cells across sessions, $\geq 30$. Altogether, this resulted in 28 mice for the familiar data and 23 mice for the novel data.

Analogous plots for what we refer to as the "Change" and "No Change" stimulus groups are shown in S3 Fig. The Change stimulus group consists of aggregate of all Hit and Miss neuronal responses, once again averaged between the time window of 150 and 750 ms immediately following an image change. The No Change stimulus group consists of neuronal responses averaged between the same time window but immediately following an image flash where the image *did not change* from the previous flash. More specifically, no change responses were

**Table 2. Shared cell and engaged trial count for behavioral data.**

| Session | Shared Cells | Hit Trials | Miss Trials | Change Trials | No Change Trials |
|---------|--------------|------------|-------------|---------------|------------------|
| F1 | 126 (84–165) | 114 (96–143) | 44 (30–53) | 161 (137–180) | 325 (267–365) |
| F2 | | 98 (64–120) | 30 (18–46) | 128 (110–150) | 258 (190–289) |
| N1 | 118 (76–192) | 121 (95–135) | 32 (26–34) | 148 (125–166) | 297 (232–324) |
| N2 | | 99 (73–137) | 32 (20–41) | 141 (104–160) | 268 (212–342) |

All numbers are: median (Q1-Q3) over $n_{mice}$ = 28 and 23 for the familiar and novel data, respectively.

collected only in (engaged) Hit and Miss trials prior to the image change, but only after at least three image flashes had occurred. This was done so the distribution of the number of image flashes prior to a neuronal response was similar between the Change and No Change data. Response vectors for these two stimulus groups are also averaged over the same 600 ms response window.

**Fig 4 details.**   Many of the plots in Fig 4 were generated analogously to those in Fig 3, so refer to Sec. 4.2 above for details. A notable exception is that, due to the comparatively large stimulus group sizes, drift data from the two groups was not averaged over as it was across the 30 stimulus groups in the passive data. The performance metrics shown in Fig 4c, 4d and 4e are only for the Hit stimulus group, see S3 Fig for other groups. Once again, linear fits on the variance explained versus drift magnitude and PC-drift angle were better using a linear variance explained scale versus logarithmic variance explained scale. The performance metrics shown in Fig 4f make use of an experimentally-defined performance metric, $d'$, averaged across all engaged trials, see Ref. [33] for additional details. Specifically,

$$d' \equiv \Phi^{-1}(R_H) - \Phi^{-1}(R_F) \tag{15}$$

where $\Phi^{-1}(\cdot)$ is the inverse cumulative Gaussian distribution function and $R_H$ and $R_F$ are the running Hit and False Alarm rates, respectively. Finally, Fig 4g and 4h were generated analogously to Fig 3e and 3f, respectively.

**S2 Fig details.**   For S2(b) Fig, variational space overlaps were computed between two stimulus groups using Eq (10). Values for randomly oriented variational spaces were analytically *approximated* to be $\Gamma_p^{s,s'} = \max(\lceil D_p^s \rceil, \lceil D_p^{s'} \rceil)/n$. To arrive at this expression, without loss of generality take the smaller of the two subspace to span the first $D$ directions of neural state space. We can construct the larger subspace, with dimension $D'$, by randomly drawing $D'$ unit vectors in neural state space sequentially, ensuring they remain orthogonal. The first of these has an average $\gamma$ of $D/n$ with respect to the smaller neuronal subspace. Since the second basis vector must be drawn from the subspace orthogonal to the first, it has an average $\gamma$ of $D/(n-1)$. This process repeats to the last basis vector, with $\gamma = D/(n - D')$. In practice, since $n \gg D'$, we neglect the numerical difference and simply approximate $\gamma$ for all $D'$ basis vectors to simply be $D/n$. Then, summing together each $\gamma$ and dividing by the $D$, we arrive at our aforementioned expression.

**4.3.1 Behavior control.**   It is well known that behavior can influence neuronal responses and their stability [15, 26, 68]. In this section, we discuss additional checks that were performed on the behavioral data to ensure that the properties of representational drift that we observe cannot be explained by the behavior of the mice. The influence of behavior on the passive data has been analyzed previously in Ref. [11], where they found behavioral metrics such as running speed and pupil area could not explain the drift they observed. Additionally, recent work has highlighted behavioral contributions to within-session drift, i.e. drift on the order of hours [25]. Notably, here we are concerned with drift on longer time scales (days) and purposely analyzes data that is averaged over entire sessions in an attempt to remove any latent effects such as behavioral dependence.

We will be specifically concerned with the behavioral metrics of running speed, pupil area, and eye position throughout this section. All three of these metrics were available for all but one mouse that was used only in the familiar analysis.

Since we are only concerned with the mice's representations during the response window of Hit and Miss trials during which they are engaged with the task (see above), we similarly filter the behavioral metrics to these subsets of time points. Additionally, behavioral metrics are averaged over the response windows in the same way as the response vectors. Altogether, this

yields a single quantity for the three behavioral metrics for each Hit and Miss trial (S7(a) Fig). Filtering to the subset of engaged response windows and also averaging within the response window yields a more narrow distribution of the behavior metrics compared to the raw data across the entire session (S7(b) Fig).

To begin, we investigated how much the behavioral metrics could explain the size of the response vector. For each mouse, we fitted the magnitude of the response vector of a given trial using a 4d linear regression (the three behavioral metrics, with eye position being two numbers) and indeed found significant evidence that said metrics influence the response vectors for some of the mice (F-test, S7(c) and S7(f) Fig). As mentioned above, the fact that behavior can influence neuronal responses is well known [15, 26, 68], but what is important for our purposes is that differences in behavior are not influencing the characteristics of drift we are observing in this work. To confirm that there is no significant difference in the behavior of mice across sessions, we trained an SVC to distinguish sessions F1 and F2 from the average behavioral metrics of the corresponding session. Specifically, we used 10-fold cross validation and further averaged over 10 shuffles of the data and found the SVC achieved chance accuracy, $47.3 \pm 2.0\%$ (mean $\pm$ s.e.). Similarly, for distinguishing N1 and N2, we found $48.9 \pm 2.2\%$ accuracy (mean $\pm$ s.e.).

To understand the differences in behavior across session, let $b_I$ be one of the four numbers representing the behavior of session $I$, we define the normalized behavior difference as

$$\frac{b_{F2} - b_{F1}}{\frac{1}{2}(|b_{F1}| + |b_{F2}|)} \, , \tag{16}$$

and defined similarly for the novel session. For running speed, pupil area, and $x$-$y$ eye position, we find the average normalized behavioral difference to be close to zero across all mice (S7(d) and S7(g) Fig). Finally, for each mouse, we can investigate if its change in behavior across sessions explains the size of the drift observed. For example, using 1d linear regression, the change in mean run speed across session does not help explain the amount of drift we observe (S7(e) and S7(h) Fig). Furthermore, fitting the drift to all four behavior metrics, we find the Hit and Miss trial drift we observe has $p = 0.315$ and $0.069$, respectively for the familiar data, and for the novel data we find $p = 0.201$ and $0.261$, respectively (4d linear regression, F-test).

**4.3.2 Additional data quality control.** In order to verify that the behavioral data can reliably be compared across distinct imaging sessions, in this section we describe several additional quality control metrics that were performed on said data. We systematically show that several variances in day-to-day collection do not explain the drift characteristics we observe in the data. In particular, this section analyzes the region of interest (ROI) masks used to identify cells within a given session and also across sessions. Each cell in each session has an associated ROI mask that consists of binary values identifying where in the imaging plane luminescence was detected in the given session. More details about the ROI masks and the data processing that went into defining them can be found in the technical whitepaper [33].

To quantify the matching of a given ROI mask between two sessions, we introduce two metrics that can be computed for each cell that is matched across sessions. Let $M^s(c)$ represent the number of nonzero values in the ROI mask of cell $c$ in session $s$. To quantify how much the area of the ROI mask changes between sessions, we define the *normalized area difference*, $A$, of cell $c$ to be

$$A^{s,s'}(c) \equiv \frac{M^{s'}(c) - M^s(c)}{M^s(c)} \, . \tag{17}$$

We will always take $s$ to be the earlier session, i.e. $s$ = F1 or N1. Note that $A$ is positive (negative) when the later (earlier) session has more pixels.

In addition to area difference, we want a metric for how similar the shape of a cell's ROI masks are between session. If we flatten the ROI masks into vectors, the normalized dot product between the resulting vectors can capture their similarity. However, we must compensate for the fact that the absolute X-Y location of a given cell in the imaging plane can change between the sessions. To this end, we scan the two masks over one another by iterating over a (pixel-valued) offset of $(x, y)$ and seeing how this changes the alignment. Let $\mathbf{m}^s_{x,y}(c)$ be the flattened ROI mask of cell $c$ in session $s$ with an offset of $(x, y)$. Define the *maximum alignment*, $\sigma$, to be

$$\sigma^{s,s'}(c) \equiv \max_{(x,y)} \frac{\mathbf{m}^s_{x,y}(c) \cdot \mathbf{m}^{s'}_{0,0}(c)}{\|\mathbf{m}^s_{x,y}(c)\|_2 \|\mathbf{m}^{s'}_{0,0}(c)\|_2} \; . \tag{18}$$

Note that we are only offsetting one mask from the other (the dot product would not change if we offset both masks the same amount). Since the components of a given $\mathbf{m}$ consist of only binary values, the maximum alignment obeys $0 \leq \sigma \leq 1$.

With metrics to quantify how well each cell is matched between sessions, we investigated whether certain regions of the imaging plane were more poorly matched between sessions than others. This could occur if, for example, the imaging plane of the later session were tilted slightly relative to its location in the earlier session. To do so, we plotted the normalized area differences of cells as a function of their location in the imaging plane (S8(a) Fig). Across all mice and for both the familiar and novel drift, we then fitted the data using a 2d linear regression and used an F-test to determine that the fitted values did not differ significantly from a flat plane (S7(k) and S8(g) Figs). This same procedure was repeated for each cell's maximum alignment between sessions (S8(b) Fig), also finding no significant evidence of a trend toward one direction in the plane (S8(h) and S8(l) Fig). We took this to mean, across all mice used in computing drift metrics, there was no significant trend of one particular region of the imaging planes being more poorly matched than other regions.

Next, we aimed to check if certain drift characteristics could be explained by how well cells were matched across sessions, as measured by the previously introduced measures. We plotted the amount of drift in a given cell (both in true value and magnitude) as a function of its normalized area difference (S8(c) and S8(d) Fig). Again, we fit this data using linear regression and found no significant evidence of drift varying as a function of this metric across mice and sessions (S8(i) and S8(m) Fig). A similar check as a function of maximum alignment also yielded no trend. Additionally, we plotted drift (again in true value and magnitude) as a function of the cell's location in the imaging plane (S8(e) and S8(f) Fig). This was meant to investigate if any particular part of the imaging plane exhibited a statistically significant amount of drift. Analyzing the 2d linear regression fit, we again found no significant evidence of drift being biased toward one particular part of the plane (S8(j) and S8(n) Fig). Of course, absence of a significant result is not a conclusive proof of absence, but beyond the lack of significance, visually there does not appear to be a consistent relation between location in the imaging plane and cell matching, location in the imaging plane and drift, and quality of cell matching from session to session and the amount of drift.

We then checked if the quality of cell matching for all matched cells belonging to a given mouse could explain the amount of drift seen in said mouse across sessions. To do so, we plotted the magnitude of drift (normalized by the number of cells imaged) as a function of both the average magnitude of normalized area difference (S9(a) and S9(c) Fig) as well as the average alignment (S9(b) and S9(d) Fig). Fitting said data using linear regression, we again found

no trend showing that the amount of drift observed, as measured by its magnitude, could be explained by the overall cell matching quality of the mouse.

Finally, we investigated if the qualitative results of drift geometry would change if we only kept cells that were "very-well-matched" between sessions. We defined a cell $c$ to be well-matched if $A^{s,s'}(c) < 0.15$ $and$ $\sigma^{s,s'}(c) > 0.85$. This criterion was quite strict and results in a large proportion of the matched cells being thrown out across mice (S9(e) and S9(i) Fig). Using the same pipeline we previously used to compute several important drift geometric characteristics, we find the very same qualitative properties are present in the drift when only the well-matched cells are included (S9(f)–S9(h) and S9(j)–S9(l) Fig).

**4.3.3 Robustness of response vector.**   To test the robustness of the mean response vectors used in this dataset, we a subset of trials and computed how much this caused the response vector to change. Let $\bar{\mathbf{x}}$ be the true mean response vector and let $\bar{\mathbf{x}}'$ be the mean response vector once *half* of the members of the stimulus group are removed. We define the change in the mean response vector to be $\Delta\bar{\mathbf{x}} \equiv \|\bar{\mathbf{x}} - \bar{\mathbf{x}}'\|_2 / \|\bar{\mathbf{x}}\|_2$. For the novel data, removing either the even or odd Hit trials yields on average $\Delta\bar{\mathbf{x}} = 0.21$ (for the familiar, $\Delta\bar{\mathbf{x}} = 0.26$). Since we found the size of the drift to be comparable size of the mean response vectors (S2(e) and S2(f) Fig), this means the instability from removing half of the trials is small compared to the size of the drift vector. As expected, since there are significantly fewer Miss trials (generally 1/4 to 1/3 the amount compared to Hit trials, see Table 2) we find $\Delta\bar{\mathbf{x}}$ to be much higher, 0.43 for both the familiar and novel data.

## 4.4 Artificial neural networks details

We train convoluational neural networks on the CIFAR-10 dataset [38]. We take the feature space to be the (post-activation) values of the penultimate fully connected layer in both setups.

Our CNNs were trained with an L2 regularization of $1 \times 10^{-3}$ and stochastic gradient descent with a constant learning rate of $\eta = 1 \times 10^{-3}$ and a momentum of 0.9. The CNN architecture used is shown in detail in S4 Fig, briefly its layers are

$$\text{2d conv, pool} \rightarrow \text{2d conv, pool} \rightarrow \text{fully connected, ReLU} \rightarrow \text{fully connected, ReLU} \rightarrow \text{linear readout}.$$

All networks were trained until a steady accuracy was achieved. Across all networks, we observed an accuracy significantly higher than guessing, within the range of 60% to 70%. Training is done using a batch size of 64.

All types of noise are redrawn every forward pass, unless otherwise stated. Additive and multiplicative node dropout are applied directly to the feature layer preactivations. Additive gradient and weight dropout are applied only to the weight layer feeding into the feature layer, i.e. the $120 \times 84$ weights. Node dropout is applied to the 84 nodes in the feature layer.

Unless otherwise stated, feature space data was collected for 10 epochs at a rate of 10 times per epoch. Features space values are collected for a test set, consisting of 1000 examples of each class. Said samples are shuffled and passed through the network over 10 forward passes, during which the noise of interest (e.g. dropout) is still applied, but notably it is still recalculated during each forward pass. This is done to collect feature space representation under several different random draws of noise, so as not to bias the feature space representations toward a particular draw of the random noise. Note the weights of the network are not updated during these forward passes, so the network is not "drifting". In practice, only feature space values for a two-class subset (frogs and birds) were saved, each consisting of 1000 examples of each class (but distributed over 10 forward passes).

For node dropout and additive Gaussian noise, to ensure our results were not dependent on the feature layer size, we trained additional architectures with both larger and smaller

feature layers (sizes 126 and 56, respectively). Although we observed minor quantitative differences, the overall qualitative features of drift we observed were unchanged in these setups. For node dropout, we also investigated "drift" in the feature space representations from simply redrawing the dropped weights. Since the exact draw of dropout noise indeed affects the feature space representation, this does result in a change in the representations. Notably however, we found that the drift is distinct from that due to dropout in a continual learning setting and does not resemble the experimental data nearly as well.

**Hyperparameter scan details.** To determine noise injection hyperparameters and generate the data shown in Fig 5a, we conducted scans over values of $\sigma$ and $p$ for each of the five types of noise. For node and weight dropout, we scanned over $p$ values from 0.1 to 0.9 (in steps of 0.1). For additive node, additive gradient, and multiplicative type noise, we scanned over $\sigma$ values of $10^{-3}$ to $10^{1}$. Each network was trained until overfitting started to occur and the feature space representations from all 10 classes of CIFAR-10 were used to determine best fit metrics to the noise.

To evaluate the fit of our CNN models to the experimental data, we look across the four following criterion:

1. The percentage of drift's magnitude that lies in variational space, and its constancy in $\Delta t$.

2. Drift's trend of lying more and more obtuse from the largest PC dimensions, as a function of $\Delta t$. Note we condition this metric on the fact that this is well fit by a linear relationship, as it is in the experimental data, penalizing models whose $r$ value is low even if the slope matches experiment well.

3. Drift's tendency to lead to a flow of variance explained out of the top PC directions. Similar to the previous metric, we also condition this on a good linear fit.

4. Angle difference between SVC classifiers and the relative classification accuracy.

To quantify these four criterion, we evaluate the following sum of $Z$-score magnitudes

$$Z_{\text{total}} = \underbrace{|Z(b_{\bar{\gamma}})| + |Z(a_{\bar{\gamma}})|}_{1.} + \underbrace{\tilde{Z}(a_{\bar{\theta}}, r_{\bar{\theta}})}_{2.} + \underbrace{\tilde{Z}(a_{\bar{\nu}}, r_{\bar{\nu}})}_{3.} + \underbrace{|Z(\Delta\theta_{\text{SVC}})| + |Z(a_{\text{rel}})|}_{4.}, \qquad (19)$$

where $Z(\cdot)$ is the $Z$-score of ANN metric relative to the experimental data, $Z(x) = (\mu_x^{\text{ANN}} - \mu_x^{\text{exp.}})/\text{s.e.}_x^{\text{exp.}}$, and $\tilde{Z}$ are the $Z$-scores of the second and third metrics conditioned on good fits,

$$\tilde{Z}(a, r) \equiv \begin{cases} 2|[Z(r)]^0| & |[Z(r)]^0| > |Z(a)|, \\ |[Z(r)]^0| + |Z(a)| & \text{otherwise}. \end{cases} \qquad (20)$$

We use $[\cdot]^0$ to denote a clipping of the $Z$-score to be at most 0, so we do not penalize models that are better fits than the experimental data. Note the $Z$-score of the slope of the linear fit only contributes if its $Z$-score is smaller than that of the $Z$-score of the $r$-value.

We compute $Z_{\text{total}}$ relative to the passive data, since that dataset has significantly more mice and thus sharper fits on all the parameters. $Z$-scores on the familiar/novel behavioral data are in general quite low and continue to be best fit by dropout-type noise, although the exact values of best fit for $p$ and $\sigma$ differ slightly.

Each of the four metrics contributes two $Z$-scores to this expression. Note we omit measuring model performance using metrics that vary significantly between the experimental datasets, for instance, the size of the drift magnitude relative means. We found that our overall

conclusion, that node and weight dropout best match the experimental data, were not sensitive if we included the aforementioned metrics to evaluate performance.

**Additional Fig 5 details.** With the exception of the hyperparameter scans, the rest of the ANN data in this figure was averaged over 10 separate initializations. Although training/testing were conducted over all 10 classes of CIFAR-10, plots here are for a single class, frogs. We found the qualitative characteristics of different classes to be the same, and the quantitative values did not vary significantly. Fig 5b and 5c were generated analogously to the $\Delta t$ fits of Fig 2f and 2h, respectively (see above). Similarly, Fig 5d, 5e and 5f (as well as their node dropout analogs) were generated analogously to Fig 3a, 3b and 3c, without the averaging over stimulus groups.

**Fig 6 details.** SVCs are trained with 10-fold cross validation using the same parameters used on the experimental data above, but now on the feature space representations of the two-class subset of CIFAR-10. Once again, the angle between SVCs is the angle between their normal vectors and cross classification accuracy is defined as in Eq (14) above. To produce Fig 6a, we computed the pairwise alignment between all SVCs within a window of 10 epochs and averaged this quantity over 10 initializations. Fig 6b used the same pairing but computed the cross classification accuracy, Eq (14), between all pairs. Note the time difference can be negative here because the data could be at a later time than when the classifier was trained. Fig 6c used the same pairing scheme as well, but instead computed the angle between the readout vectors for each class, and then averaged the quantities across classes. For Fig 6d, for a given class, we computed the stimulus group's drift relative to the direction of its corresponding readout vector. The drift of a stimulus group compared to the other group's readout looked similar. The chance percentage was computed by generating two random vectors in a space the same as as the feature space ($n = 84$) and computing their average deviation from perpendicular. Our previous findings indicate that the direction of drift is far from completely random in neural state space. However, they are not inconsistent with drift occurring in a high-dimensional space. The chance percentage shown in Fig 6d changes little if we restrict the drift direction to be constrained along many directions.

The targeted node dropout results in Fig 6e as well as in S4 Fig were generated by preferentially targeted particular nodes during dropout. Let $P_\mu$ for $\mu = 1, \ldots, n$ be the probability of dropping a particular node in the feature layer during each forward pass. For the regular node dropout in this figure, which is identical to the node dropout data in Fig 5, we simply have $P_\mu = p = 0.5$ for all $\mu = 1, \ldots, n$. To compute targeted dropout, 10 times per epoch we collected the feature space representations of the test set. Across the test set, we compute the variance for each node and use this to compute the ratio to total variance for each node, $0 \leq v_\mu \leq 1$ (this is similar to the ratio of variance explained of PC dimensions, $v_i$). Using this ratio of total variance for each node, 10 times per epoch we update the $P_\mu$. For targeting nodes of maximum variance, we use the expression

$$P_\mu = [A v_\mu]_0^1, \tag{21}$$

where $A \geq 0$ controls the strength of the dropout and $[\cdot]_0^1$ clips the value between 0 and 1. For $A = 1$, on average only a single node is dropped, since $\sum_\mu v_\mu = 1$. Meanwhile, to target the nodes of minimum variance, we used

$$P_\mu = \left[ A \frac{(v_\mu + \epsilon)^{-1}}{\sum_{v=1}^n (v_v + \epsilon)^{-1}} \right]_0^1, \tag{22}$$

where $\epsilon = 10^{-5}$ sets a lower threshold for the smallest variance ratio. For the aforementioned

figures, $A = 20$ and 42 for the maximum and minimum targeted node dropout data, respectively. This results in an average number of nodes dropped of $\sim$17 per forward pass for the maximum targeted nodes and 42 for the minimum node targeting (note the latter of these quantities is the same number that would be dropped under regular node dropout with $p = 0.5$, since $pn = 42$). Targeted node results were averaged over 5 initializations.

   To generate the plot in Fig 6f, we lowered the frequency of the recalculation of what nodes are dropped to once per epoch (down from once for every forward pass). All other training parameters of the network are identical to the node dropout case in the main text. Since we sample the feature space 10 times per epoch, this means the sampling rate of the feature space is smaller than the noise update scale. This data was computed over 5 network initializations.

## Supporting information

**S1 Fig. Passive data: Additional results and correlation plots. (a)** Variance explained versus mean value along the corresponding PC direction, normalized by L2 magnitude of the mean response vector. Note all mean values are positive by definition of the PC's direction. **[b-d]** Plots equivalent to Fig 5c, 5d and 5e, but instead of angle as a measure of similarity, here we are using Pearson's correlation coefficient, Eq (7). These plots are provided to show a similarity of angle between response vectors (used in the main text) with the metric used in Ref. [11]. **(b)** Correlation within-session and across the first five movie repeats between response vectors. **(c)** Correlation between mean response vectors across the three session. **(d)** Correlation as a function of time between sessions, $\Delta t$. Again, we note this decrease is quite small as a function of time ($< .01$/day). **[e-f]** SVC metrics as a function of time difference between sessions. These plots are analogous to Fig 3e and 3f, just plotted as a function of $\Delta t$ instead of by session number. Once again, note both of these metrics are relatively stable as a function of the time difference. **(e)** Angle between SVC classifiers. **(f)** Relative cross classsification accuracy. Note the time difference here can go negative beause the classifier could be trained on a sesssion earlier than the data it is tested on (and unlike the angle between classifiers, in general $a_{\mathrm{rel.}}(-t) \neq a_{\mathrm{rel.}}(t)$). (JPG)

**S2 Fig. Behavior data: Additional results and drift metrics as a function of $\Delta t$. (a)** Variance explained versus mean value along corresponding PC direction, normalized by L2 magnitude of mean response vector. Note all mean values are positive by definition of the PC's direction. **(b)** Variational space overlap, $\Gamma$ of Eq (10), between the Hit and Miss stimulus groups for each session. Pink/yellow light dots show raw data, pink/yellow dark dots/lines show average $\pm$ s.e. Grey dots/lines show average $\pm$ s.e. of two randomly oriented variational spaces of same dimensions. Notably, the two stimulus groups overlap significantly more than chance. **(c)** Magnitude of drift as a function of the earlier session's variance explained, but unlike Fig 4c the magnitude is not normalized by the full L2 magnitude of the drift. This plot is meant to show the comparatively larger drift of the novel sessions (darker dots/lines) relative to the familiar sessions (lighter dots/lines). **[d-k]** Various metrics as a function of the time between earlier and later session, $\Delta t$, for both Hit and Miss trials, across all mice. The middle and bottom rows correspond to the familiar and novel data, respectively. Colored curves are linear regression fits with the shaded region showing all fits within the 95% confidence intervals of the slope and intercept. Darker/lighter colored lines/points correspond to Hit and Miss trials, respectively. Dotted lines are best fits for class-averaged data. **(d, h)** Average angle between response vectors (Methods). **(e, i)** Average magnitude of drift relative to magnitude of mean response vector. **(f, j)** Average *ratio* of participation ratio from later session to earlier session. Note this differs from the analogous plot for the passive and ANN data in that we are taking the ratio rather than the difference. We found ratio to be a better measure given the significant

variance in size of variational/neural state spaces for the behavioral data. **(g, k)** Average drift magnitude within earlier session's variational space, ratio relative to full drift magnitude. (JPG)

**S3 Fig. Behavioral data: Miss, Change, and No Change stimulus group geometries and Change versus No Change classification.** **[a-i]** Various drift metrics of Miss (first row), Change (second row), and No Change (third row) trials and their dependence on PC dimension of the earlier session's variational space. Dark colors correspond to drift between familiar sessions, while lighter colors are those between novel sessions. Metrics are plotted as a function of the stimulus group PC's ratio of variance explained, $v_i$. Colored curves are linear regression fits. **(a, d, g)** Magnitude of drift along PC direction relative to full (L2) magnitude of drift. **(b, e, h)** Angle of drift with respect to PC direction. **(c, f, i)** Post-drift variance explained along PC direction (dotted line is equality). Linear regression fit to log(var. exp.). **[j-k]** Various SVC metrics as a function of session(s) for the Change/No Change stimulus groups. Dark solid dots/lines show average values with ± s.e. Light colored dots/lines show raw mice data. **(j)** Angle between SVC normal vectors. **(k)** Relative cross classification accuracy, Eq (14), as a function of test data session and trained data session. (JPG)

**S4 Fig. Artificial neural networks: Additional results.** **(a)** Visualization of convolutional neural network architecture. **(b)** Accuracy as a function of epoch for SGD and all five types of noise. Note, for each of the six plots, $t = 0$ is chosen to be the point where the network has relatively steady accuracy, and does not necessarily correspond to the same epoch across noise types. **(c)** Average change in the variational space dimension, $D$, from later session to earlier session as a function of $\Delta t$. **[d-g]** Additional results for node dropout, with $p = 0.5$. **(d)** Variance explained versus mean value along corresponding PC direction, normalized by L2 magnitude of mean response vector. **(e)** Variational space overlap, $\Gamma$ of Eq (10), between the two stimulus groups (frogs and birds) as a function of time (after steady accuracy is achieved) for node dropout with $p = 0.5$. Light pink dots show raw data for each trial, dark line/fill is mean ± s.e. Grey dots/lines show raw data and mean ± s.e. of two randomly oriented variational spaces of same dimensions. **(f)** Variational space overlap as a function of time difference. Green shows $\Gamma$ between same stimulus group, blue shows between two different stimulus groups, yellow shows chance percentage. Between the two classes, overlap is largest for $\Delta t = 0$, before reaching a steady value that is still larger than chance. **(g)** Angle between drift vectors as a function of time difference between initial sessions. Green shows between same stimulus group, blue shows between two different groups. **[h-j]** Various plots for targeted node dropout. Regular node dropout, with $p = 0.5$, is plotted in purple. Maximum and minimum variance targeted node dropout, Eqs (21) and (22), are plotted in pink and yellow, respectively. **(h)** Average magnitude of drift relative to magnitude of mean response vector as a function of $\Delta t$. **(i)** As a function of the variance explained of the earlier session, magnitude of drift along corresponding PC direction, normalized by full (L2) magnitude of drift. **(j)** Post-drift variance explained along PC directions, plotted as a function of the earlier session's corresponding variance explained. **[k-m]** Various plots for node dropout applied to other areas of the network. With "FC1" the fully connected layer before the feature layer (S4 Fig (a)), in purple, red, and teal, we have node dropout on only the feature layer (as in main text), both FC1 and feature layer, and FC1 only, respectively. All plots are averaged over at least 5 initializations. **(k)** Magnitude of drift as a function of $\Delta t$. **(l)** PC-drift angle as a function of variance explained. **(m)** Variance explained after drift as a function of variance explained of earlier session. (JPG)

**S5 Fig. Artificial neural networks: Geometry of drift for various types of noise.** Various drift metrics and their dependence on PC dimension of the earlier session's variational space. Metrics are plotted as a function of stimulus group PC's ratio of variance explained, $v_i$. Colored curves are linear regression fits. Grey curves are behavioral data fits from the novel sessions shown in Fig 4c, 4d and 4e. Noise hyperparameters are chosen to be best fits to experimental data across hyperparameter scans, see Fig 5a. Note equivalent plots for additive node and node dropout are in Fig 5 of the main text. **(First column)** Magnitude of drift along PC direction relative to full (L2) magnitude of drift. **(Second column)** Angle of drift with respect to PC direction. **(Third column)** Post-drift variance explained along PC direction (dotted line is equality). Linear regression fit to log(var. exp.). **(First row)** Fits for only SGD and all five types of noise. Insets show $r$-values of fits. **(Second row)** Only SGD. **(Third row)** Additive gradient noise with $\sigma = 3.0$. **(Fourth row)** Weight dropout with $p = 0.6$. **(Fifth row)** Multiplicative node with $\sigma = 1.0$.
(JPG)

**S6 Fig. Artificial neural networks: Drift geometry plotted as a function of PC dimension.** Various drift metrics and their dependence on PC dimension of the earlier session, rather than its's variational space, which is used throughout the main text. Solid dark line and shading represent mean ± s.e. for each PC. Raw data is scattered behind as points. **(First and third column)** Magnitude of drift along PC direction relative to full (L2) magnitude of drift. **(Second and fourth column)** Angle of drift with respect to PC direction. Noise hyperparameters are chosen to be best fits to experimental data across hyperparameter scans, see Fig 5a. **[a-b]** Passive data. **[c-d]** Behavioral Hit data, with darker/lighter lines/shading the familiar and novel sessions, respectively. Note raw data is not scattered for clarity. **[e-f]** ANN with only noise due to SGD. **[g-h]** Additive node with $\sigma = 0.1$. **[i-j]** Additive gradient with $\sigma = 3.0$. **[k-l]** Node dropout with $p = 0.5$. **[m-n]** Weight dropout with $p = 0.6$. **[o-p]** Multiplicative node with $\sigma = 1.0$.
(JPG)

**S7 Fig. Behavioral data: Behavior quality control.** **[a,b]** Behavior data from an exemplar mouse (specifically from a familiar session). **(a)** Run speed versus time for a sample time slice. Light purple shows raw data, dark purple points shows data during engaged Hit or Miss trials, and orange stars show average run speed for each Hit or Miss trial. **(b)** Distribution of run speeds over entire session. Purple shows raw data, orange shows average over each Hit or Miss trial. **[c-e]** Familiar sessions behavior control. **(c)** Histogram of p-values from F-test to see if the four behavioral metrics (run speed, pupil area, $x$ and $y$ eye position) could explain size of response vectors, across all mice used in familiar data. Red and blue shown Hit and Miss stimulus groups, respectively. Two dotted vertical lines mark the 0.01 and 0.05 significance levels. These are the p-values for individual tests, such that one out of 20 (100) are expected to below 0.05 (0.01) by chance. **(d)** Normalized behavioral differences, see Eq (16), between F1 and F2 sessions for run speed (RS), pupil area (PA), and $x$ and $y$ eye position (EX, EY). Large dots are means across all mice, with individual mice data scatter behind. **(e)** Scatter plots of normalized drift magnitude, $\|\mathbf{d}\|_2/\sqrt{n}$, as a function of change in mean run speed for mice used in familiar data. Dark lines show linear fits to data and respective p-values of Wald tests to see if slopes are significantly different from flat are shown. Red and blue shown Hit and Miss stimulus groups, respectively. **[f-h]** Novel sessions behavior control. **(f)** Same as (c), for novel sessions. **(g)** Same as (d), for N1 and N2 sessions. **(h)** Same as (e), for novel sessions.
(JPG)

**S8 Fig. Behavioral data: Quality control checks on cell matching.** **[a-f]** Results from an exemplar mouse's drift (specifically from novel Hit population drift when relevant). **(a)**

Normalized area difference of ROI masks, Eq (17), as a function of the later session's X-Y locations in the imaging plane. P-value shown is from F-test to see if planar fit to data is significantly different from flat. **(b)** Maximum alignment between ROI masks, Eq (18), as a function of the later session's X-Y location. P-value from same type of test used in (a). **(c)** Scatter plot of each cell's drift as a function of the cell's normalized area difference. Dark line is a linear fit with shaded area representing mean ±2 s.e., p-value from Wald Test to see if slope of fit is significantly different from flat. **(d)** Same as (c), but magnitude of a cell's drift. **(e)** Each cell's drift as a function of the later session's X-Y locations. P-value from same type of test used in (a). **(f)** Same as (e), but magnitude of a cell's drift. **[g-j]** Histogram of p-values of the same tests shown in (a), (b), (c), and (e), respectively, across all mice used in familiar data. Two dotted vertical lines mark the 0.01 and 0.05 significance levels. These are the p-values for individual tests, such that one out of 20 (100) are expected to below 0.05 (0.01) by chance. **(g, h)** Dark values represent data from single-plane imaging while lighter bars are individual imaging planes from multi-plane data (data is stacked). **(i, j)** Orange and teal bar plots (not stacked) represent data from the Hit and Miss population drifts, respectively. **[k-n]** Same as [g-j], but for the all mice used in the novel data.
(JPG)

**S9 Fig. Behavioral data: Quality control checks on cell matching. [a-f]** Results from an exemplar mouse's drift (specifically from novel Hit population drift when relevant). **(a)** Normalized area difference of ROI masks, Eq (17), as a function of the later session's X-Y locations in the imaging plane. P-value shown is from F-test to see if planar fit to data is significantly different from flat. **(b)** Maximum alignment between ROI masks, Eq (18), as a function of the later session's X-Y location. P-value from same type of test used in (a). **(c)** Scatter plot of each cell's drift as a function of the cell's normalized area difference. Dark line is a linear fit with shaded area representing mean ±2 s.e., p-value from Wald Test to see if slope of fit is significantly different from flat. **(d)** Same as (c), but magnitude of a cell's drift. **(e)** Each cell's drift as a function of the later session's X-Y locations. P-value from same type of test used in (a). **(f)** Same as (e), but magnitude of a cell's drift. **[g-j]** Histogram of p-values of the same tests shown in (a), (b), (c), and (e), respectively, across all mice used in familiar data. Two dotted vertical lines mark the 0.01 and 0.05 significance levels. These are the p-values for individual tests, such that one out of 20 (100) are expected to below 0.05 (0.01) by chance. **(g, h)** Dark values represent data from single-plane imaging while lighter bars are individual imaging planes from multi-plane data (data is stacked). **(i, j)** Orange and teal bar plots (not stacked) represent data from the Hit and Miss population drifts, respectively. **[k-n]** Same as [g-j], but for the all mice used in the novel data.
(JPG)

**S1 Appendix. Variational space overlap rotational invariance.** Here we argue that $\Gamma_p^{s,s'}$, defined in Eq (10), is invariant with respect to rotations of the orthonormal bases used to span the variational spaces of $s$ and $s'$.
(PDF)

## Acknowledgments

We thank Stefan Berteau and Dana Mastrovito for feedback on this paper. We also wish to thank the Allen Institute for Brain Science founder, Paul G. Allen, for his vision, encouragement, and support.

## Author Contributions

**Conceptualization:** Kyle Aitken, Stefan Mihalas.

**Data curation:** Marina Garrett, Shawn Olsen.

**Formal analysis:** Kyle Aitken.

**Funding acquisition:** Shawn Olsen, Stefan Mihalas.

**Investigation:** Kyle Aitken.

**Methodology:** Kyle Aitken.

**Project administration:** Stefan Mihalas.

**Resources:** Marina Garrett, Shawn Olsen, Stefan Mihalas.

**Software:** Kyle Aitken, Marina Garrett, Shawn Olsen.

**Supervision:** Stefan Mihalas.

**Validation:** Kyle Aitken.

**Visualization:** Kyle Aitken.

**Writing – original draft:** Kyle Aitken.

**Writing – review & editing:** Kyle Aitken, Marina Garrett, Stefan Mihalas.

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
