## [Decision Letter · Decision Letter 0]

25 Jul 2022

Dear Dr. Aitken,

Thank you very much for submitting your manuscript "The Geometry of Representational Drift in Natural and Artificial Neural Networks" for consideration at PLOS Computational Biology.

As with all papers reviewed by the journal, your manuscript was reviewed by members of the editorial board and by several independent reviewers. In light of the reviews (below this email), we would like to invite the resubmission of a significantly-revised version that takes into account the reviewers' comments.

We cannot make any decision about publication until we have seen the revised manuscript and your response to the reviewers' comments. Your revised manuscript is also likely to be sent to reviewers for further evaluation.

Sincerely,

Peter E. Latham

Associate Editor

PLOS Computational Biology

Thomas Serre

Deputy Editor

PLOS Computational Biology

Reviewer's Responses to Questions

**Comments to the Authors:**

Reviewer #1: Authors define geometry of neuronal representations to better understand how they drift. Authors find that neural representations change the most in dimensions with highest variance. This geometry of drift in the visual cortex appears to be similar to drift in ANNs with dropout. The authors conclude with a hypothesis that drift could help prevent overfitting. It is interesting that the rate of drift was similar across familiar and novel stimulus presentations.

Representational drift is a feature of neural data that was recently reported in the hippocampus and cerebral cortex. This feature must be quantified and characterized in order to better develop an understanding for potential roles in neural computation. Therefore, the goals of this work are interesting and important. However, authors' methods of characterization are limited in providing further insight beyond what has already been reported. Importantly, there’s little evidence for representational drift happening at all in this work. More careful comparison across sessions 1 and 2 compared to sessions 1 and 3 will be crucial to verify that representational drift is measurable in this data. Additionally, the contribution of changes in behavior across sessions was not addressed. While previous work has shown that behavioral confounds don’t explain all neural change associated with drift, it is important to examine the contribution of behavioral change in any careful analysis of neural data. The dataset analyzed in this work contains behavioral information that could be analyzed for a revised draft. Extensive work must be done to demonstrate accumulation of change over time associated with drift, and also robustness controls for the mean response vector. These revisions are possible given the previously collected data; therefore, major revisions should be required before accepting this work for publication.

Major Concerns:

*** Based on previous work, we might expect the magnitude of drift to increase as the gap between sessions increases. I couldn’t find evidence of this anywhere in the paper beyond an example figure in 2c and d. In line 168, authors state that an SVC can distinguish separate sessions. Does decoder performance improve with a larger session gap? (1-3 should have higher performance than 1-2 or 2-3)? This does not appear to be the case (or is very weakly present) in Figure 2e across different stimulus groups. How does the same plot look for the same stimulus groups (summary figures for 2c and d)? How does the rate of change in 2e compare to previous studies (ie. Xia J, Marks TD, Goard MJ, et al. (2021) etc)? More direct comparisons to previous work would be helpful.

**The authors seem to have missed an important line of work that is highly relevant to their findings from Denise Cai’s group. This work shows that there is an ‘excitable pool’ of cells that varies over days. The neurons that are most active are part of the excitable pool on any given day. As the excitable pool changes across days, neurons that were most active, are no longer active on subsequent days. These findings would result in a phenomenon similar to dropout. Authors should reference this work extensively throughout (Cai 2016, Mau 2020, etc).

** Importantly, behavioral measures were not included at all in this entire work. The available datasets include metrics like running speed, eye position and pupil area. These variables should be analyzed as extensively as the neural data and included in all session comparisons. For example, in all decoding analyses, is it possible to decode which session a dataset came from based on running speed alone?

* Line 170 : Authors cite representational drift work in hippocampus [11] as not being linked to behavioral features and claim that this is reasoning to not analyze such data in visual cortex. This explanation is not sufficient, especially given several works citing the relevance of behavioral features when analyzing neural data (Musall et al 2019, Clopath et al 2017, Liberti 2022).

All comparisons between odd and even trials should also include a comparison to the first and second half of sessions in order to determine whether there was any drift recorded over a single session (ex. in line 169 and 282). Additionally 3e,f should have odd even/ first half second have same day comparisons.

Line 9: The idea that neurons represent external stimuli is a hypothesis and should be stated as such. (Brette, 2018; Freeman and Skarda, 1990) The authors talk a lot about representations but make no effort to link neural activity to any particular representation. Given (Musall et al 2019) it will be important to determine which external stimuli and behavioral variables are contributing to neuronal activity.

Line 248: Authors state that decoding accuracies are higher than in previous work [8,9]. However, my understanding is that the timescale for this work is much shorter and it is particularly difficult to compare results to [9], where authors performed an 8-way classification. What is the actual accuracy? The results in the present study are obfuscated by plotting relative accuracy. A more direct comparison to previous work would be more informative.

Authors claim that the number of days between sessions is only weakly correlated with drift starting at line 179. I don’t see any comparison between the number of days between sessions compared to the number of sessions, as was reported in [9]. The authors should do a more thorough analysis as in figure 4d of ref [9] before making these claims.

When reporting low dimensional subspaces, authors should cite other work that has reported similar properties (Gao and Ganguli, 2015; Mastrogiuseppe and Ostojic, 2018).

Concerns about mean response vector analysis

**The authors’ main findings are largely dependent on analyses of the mean response vector. This vector is calculated by averaging the activity of neurons across 600ms-1s of imaging data. Authors don’t provide reasoning for this choice in window size. I’m curious how variable the mean response vector is over the course of this time. Authors should quantify this movement by using a sliding window and comparing the movement over time within a 1 second interval to across session movement. I’m especially concerned about the robustness of this vector. If you hold out a subset of trials (odd/even or first half/last half) and calculate the mean response vector within one session, how much does it change? How does that compare across sessions? If this change doesn’t accumulate across sessions, I wouldn’t refer to this phenomenon as ‘representational drift’.

Does the rate of change of the mean response vector change over the course of the imaging period? I would like to see a quantification of 2c for all mice. For example, it appears as though repeats 4 and 5 are more dissimilar than 1 and 2.

Line 158, 395 The authors often report that mean is correlated with variance explained, but doesn’t this have to be true for neural activity (which doesn’t go below zero)?

Include experimental data for comparison in 5d-i, 6.

Minor Concerns:

Line 30: Also provide examples of stability in connectivity (Holtmaat and Svoboda, Nat Review Neuro 2009)

Line 38: Not all refs in 3-12 show representational drift. Remove [4,6,7] from this citation.

Line 38 Include recent work on bats as a counter example (Liberti 2022)

Line 55 : Define ‘magnitude of neuronal representations’

Line 90: Drift is across days to weeks, not seconds. Is there a citation for ‘seconds?’

Lin 99: Include citation to [Michael E Rule et al (2020) Stable task information from an unstable neural population.]

Line 43: Central pattern generator doesn’t encode a representation. It generates muscle patterns. This is a confusing reference to [15]. This system is robust to differences in neural activity. Maybe just change the wording here.

figure 1 legend: ‘PCA decomposition’ is non standard language. Change to ‘we perform PCA’

Line 293: Unclear what a ‘group’ is

298 previously

343 What are network dynamics in a feedforward network?

474 Sparsity is different than low dimensionality

486 What is the reasoning for sparsity to target active neurons rather than less active?

540 Citations or examples for more realistic network models?

582 tells

604 Natural

663 typo? I couldn’t parse.

Reviewer #2: ===============

Summary

The paper studies representation drift using neural recordings and artificial network dynamics. The former shows the type of geometrical changes in neural activity, the latter tests different types of noise in order to reproduce those geometrical changes. The authors find that node dropout is the most likely candidate. The paper is well-written and, to my knowledge, the results are novel. Moreover, the suggested link between representation drift can facilitate both experiments (as the results suggest a very specific type of noise) and theoretical studies of how noise affects neural representations.

Questions

Neural data question: as you have access to neural recordings, wouldn’t it be possible to check for node dropout? You’ve tested relatively high dropout rates (>10%), which should be detectable from data.

I think the deep learning side of the paper can be significantly improved.

First, the networks chosen in experiments are not trained to convergence (as you run SGD with a constant, rather than a decreasing, learning rate). If you train to convergence, your network might become more sensitive to noise during the second phase of learning, and potentially change the results. (Note that this is based on my intuition and may not be true. However, it’s always good to know if your results can be affected by tweaks in the learning procedure.)

Second, it would be good to know your results are not sensitive to the number of feature neurons. You can try scaling the whole network, or just the feature layer (e.g. 2x, 4x) to see if the changes are consistent across network sizes. Scaling the whole network would also improve CIFAR10 performance (the current 60-70% is rather far from the standard 85-90% you get with a ResNet18), although I don’t think the exact accuracy is important for this study.

Third, are your results specific to continual learning? You can train the network as you did, and then run the same experiments with re-drawn dropout noise, but without training. This setup would be somewhat closer to the passive data.

Minor comments

Lines 44-46: citations for both sentences would be helpful.

164: 5-fold CV on what subsets of data?

242: citation [8] should be explained rather than just stated.

287: the sentence under (1) reads weirdly, maybe “a drift magnitude of the same order as the mean response vector” would be correct?

360-361: “both” refers to three options in the sentence.

494: change”s”

508: that’s not quite true. ResNet, arguably the most popular architecture, doesn’t use dropout. Changing “almost all” to “many” would make the statement more accurate.

509-510: citation needed, even though it’s more or less common knowledge.

Fig. 5 (and similar figures): why don’t the plots reach 1 on the x axis (variance explained)? I’m assuming it’s because your discarded very small eigenvalues (as mentioned in Methods), but Fig. 5 plots stop surprisingly far from variance explained = 1.

Reviewer #3: Aitken et al. present a study of representational drift by analysis of experimental recordings in the mouse visual cortex (Allen Brain Observatory data) and simulations of artificial neural network (ANN). The focus of the study is to investigate the geometrical means by which the neural space is transformed in the process of drift. The data is presented in detail and from many different aspects, and some properties of the drift shown here are interesting and thought provoking. Additionally, the finding that the node-dropout method in the ANN can replicate many features of experimental drift is very interesting. However, there are some ambiguities and apparent contradictions in the results that require further explanations (see below).

1- The observation that there is a weak dependence of the drift on the time difference between the sessions is not very clear. In Fig. 2e, the small slope of the plot is understandable but the large intercept (around 60 degrees) makes the result counterintuitive. The plot suggests that, irrespective of the days between the sessions, there is always around 60 degrees angle difference between the mean response vectors for a given group. In other words, it appears as if the drift happens abruptly as a function of time and then stays constant. This is in apparent contradiction with the general notion that drift is a gradual and ongoing process as reported in previous studies. Specifically, in the study of Ref [11] that is based on the same data, a gradual continuous drift was found. Please explain.

2- The findings that there is a decrease of later variance explained along the top PC directions (of the previous session) and an increase in later variance explained for lower PC directions are interesting. However, a more complete geometrical intuition could be provided here. The authors explain that the variance flows out of certain dimensions to others. Is that also happening with respect to the PC directions of the current session? If that’s the case, the D metric (variational space dimension) should decrease from an earlier to a later session which does not seem to be the case from Fig 2g. Please clarify. As a possibility, a rotation of the data from one session to another could explain the increase and the decrease trends noted above while D stays fixed. Please explain if that could be compatible with the data.

3- In this study, the variational space plays a central role in characterization of the drift. A natural question is to what extent the defined variational space represents the drift itself in the short term. Given that Ref [11] also found that drift occurs within minutes in the same data, it could be a possibility that at least part of the variation we observe within different repeats of a given block in a session is due to the drift. In that case, it is not surprising that the between session drift is primarily along the top PC directions. Did the authors see any trend within the variational space (10 data points) that suggests there is a within-session drift? If that’s not the case, how do they explain the observed relationship between the drift magnitude and the variation explained along PC directions?

4- For the ANN, please specify when exactly the noise was injected in the process of training. (i.e., for each noise type, did you first train the network without noise until steady accuracy was achieved, and then started injecting the noise from epochs 0-10?) This is not very clear upon reading the text and could be added, for example, to the paragraph ending in line 346.

5- In the paragraph starting in line 372, please clarify how the variational space was measured/defined for the ANN. I assume that this was measured for the 1000 samples of each class. Was this measured at epoch 0, after a steady accuracy was achieved with training that included noise?

Minor comments:

6- Referring to line 157, what is the actual (average) value of D? Since we know it is always less than 10, it would be meaningful to report its actual value as well.

7- Please clarify if the pairwise angle plots in Fig. 2c are from one session only. Also, it appears from the plots that certain blocks (almost the first half) are more stable across repeats, while the other half are relatively unstable (i.e., part of the diagonal line in the plots between the consecutive repeats disappear). Please explain what aspect of data or experimental design could cause such effect.

8- In the last paragraph of page 18 (two lines after line 580), by “projection vector” the authors perhaps meant “projection matrix” or “projection operator”.

**Have the authors made all data and (if applicable) computational code underlying the findings in their manuscript fully available?**

Reviewer #1: Yes

Reviewer #2: Yes

Reviewer #3: **No: **The experimental data used in the study is publicly available. I am not aware if the computational codes are available.

PLOS authors have the option to publish the peer review history of their article (what does this mean?). If published, this will include your full peer review and any attached files.

Reviewer #1: No

Reviewer #2: No

Reviewer #3: No
---

## [Decision Letter · Decision Letter 1]

20 Oct 2022

Dear Dr. Aitken,

Thank you very much for submitting your manuscript "The Geometry of Representational Drift in Natural and Artificial Neural Networks" for consideration at PLOS Computational Biology.

As with all papers reviewed by the journal, your manuscript was reviewed by members of the editorial board and by several independent reviewers. In light of the reviews (below this email), we would like to invite the resubmission of a significantly-revised version that takes into account the reviewers' comments.

We cannot make any decision about publication until we have seen the revised manuscript and your response to the reviewers' comments. Your revised manuscript is also likely to be sent to reviewers for further evaluation.

Sincerely,

Peter E. Latham

Academic Editor

PLOS Computational Biology

Thomas Serre

Section Editor

PLOS Computational Biology

Reviewer's Responses to Questions

**Comments to the Authors:**

Reviewer #1: I would like to thank the authors for their extensive work addressing my concerns and the concerns of the other authors. However, two major concerns remain, preventing me from recommending this work for publication in its current form.

The authors claim that, “there does not seem to be a consensus that drift continues to “accumulate” over time.” This statement is simply not true. Drift is defined as gradual changes in population responses that accumulate over time(Rule et al. 2019; Deitch et al. 2021; Schoonover et al. 2021; Driscoll et al. 2022). Instead, authors find a similar amount of change between sessions 1-2 and 1-3. Without any evidence of gradual accumulation of change in this work, I don’t think it’s appropriate to use the term ‘representational drift’ throughout. Based on comparisons to Deitch et al. 2021, it seems as though the time frame of this work is not long enough to see changes associated with drift.

I suggested the authors should do a more thorough analysis of behavioral contributions to the geometrical changes that they report. I was not satisfied with the response, especially given recent work showing behavioral contributions to drift(Sadeh and Clopath 2022). I’m confused and surprised that the authors claim that behavior has little to no contribution on representational drift, especially given the recent analysis of the same dataset. Maybe this has some relation to subsampling the data such that only the most similar trials are analyzed? More direct comparisons to analyses in (Sadeh and Clopath 2022) will be critical.

Given these major concerns, I do not think this work should be published in its current form.

Reviewer #2: I think the authors adequately addressed my concerns, and I think the added results/discussions improve the paper. I vote for accept.

Reviewer #3: From the authors’ responses it is clear that the smallest timescale at which the drift is measured in this study is much larger than that in previous studies, and this may be a main reason why some results are in apparent contradiction to those from other studies (for example that the effects don’t seem to “accumulate” over time). The analogy of random-walk on a sphere is helpful and is also compatible with some previous studies. In the new text in the Discussion, an explanation is provided in the paragraph starting at line 577. It would be more beneficial to the reader to emphasize this distinction more clearly (i.e. that the change in the response vector is being measured in the “stabilized” regime). I suggest to clarify this also in the Introduction or when the results are presented, as this underlies many findings in the study.

Did the authors find any evidence of a short-term/within-session drift from analyses other than SVC? If short-term drift cannot be reliably verified or measured, this should be mentioned in the paper. Please also compare to the study of Ref [11] in the same data that found within minutes drift. Along these lines, please also mention in the main text (e.g. in the Discussion) that the variational space may contain, to some extent, the within session drift, as was explained in response to my third question.

**Have the authors made all data and (if applicable) computational code underlying the findings in their manuscript fully available?**

Reviewer #1: Yes

Reviewer #2: Yes

Reviewer #3: Yes

PLOS authors have the option to publish the peer review history of their article (what does this mean?). If published, this will include your full peer review and any attached files.

Reviewer #1: No

Reviewer #2: No

Reviewer #3: No
---

## [Decision Letter · Decision Letter 2]

7 Nov 2022

Dear Dr. Aitken,

We are pleased to inform you that your manuscript 'The Geometry of Representational Drift in Natural and Artificial Neural Networks' has been provisionally accepted for publication in PLOS Computational Biology.

Best regards,

Peter E. Latham

Academic Editor

PLOS Computational Biology

Thomas Serre

Section Editor

PLOS Computational Biology

Reviewer's Responses to Questions

**Comments to the Authors:**

Reviewer #1: Thank you for explaining differences in the behavioral results from Sadeh and Clopath. It makes sense that the timescale would contribute to this difference. I still worry that the timescale of change that you study is misleading about drift and will lead to confusion around the term in the field. I would feel better if you changed line 50 to say,

Many recent studies have confirmed that, under persistent performance, neuronal encodings undergo “representational drift”, i.e. a gradual accumulation of change in the representation of certain information over weeks [6–12] (though see Refs. [13–15] for counterexamples).

I know you use the Deitch study as a reference for 'minutes', but I worry those shorter timescale changes are due to behavioral change, as shown in Sadeh and Clopath.

**Have the authors made all data and (if applicable) computational code underlying the findings in their manuscript fully available?**

Reviewer #1: Yes

PLOS authors have the option to publish the peer review history of their article (what does this mean?). If published, this will include your full peer review and any attached files.

Reviewer #1: No

---

## [Editor Report · Acceptance letter]

18 Nov 2022

PCOMPBIOL-D-22-00835R2 

The Geometry of Representational Drift in Natural and Artificial Neural Networks

Dear Dr Aitken,

I am pleased to inform you that your manuscript has been formally accepted for publication in PLOS Computational Biology. Your manuscript is now with our production department and you will be notified of the publication date in due course.

With kind regards,

Anita Estes
